# SEMPO: Lightweight Foundation Models for Time Series Forecasting

**Hui He[1,2]**\* **Kun Yi[3], Yuanchi Ma[1], Qi Zhang[4], Zhendong Niu[1]**† **Guansong Pang[2]**†

[1]Beijing Institute of Technology, [2]Singapore Management University,
[3]State Information Center, [4]Tongji University
{hehui617, yma, zniu}@bit.edu.cn, kunyi.cn@gmail.com
zhangqi_cs@tongji.edu.cn, gspang@smu.edu.sg

## Abstract

The recent boom of large pre-trained models witnesses remarkable success in developing foundation models (FMs) for time series forecasting. Despite impressive performance across diverse downstream forecasting tasks, existing time series FMs possess massive network architectures and require substantial pre-training on large-scale datasets, which significantly hinders their deployment in resource-constrained environments. In response to this growing tension between versatility and affordability, we propose **SEMPO**, a novel lightweight foundation model that requires pretraining on relatively small-scale data, yet exhibits strong general time series forecasting. Concretely, SEMPO comprises two key modules: 1) *energy-aware SpEctral decomposition module*, that substantially improves the utilization of pre-training data by modeling not only the high-energy frequency signals but also the low-energy yet informative frequency signals that are ignored in current methods; and 2) *Mixture-of-PrOmpts enabled Transformer*, that learns heterogeneous temporal patterns through small dataset-specific prompts and adaptively routes time series tokens to prompt-based experts for parameter-efficient model adaptation across different datasets and domains. Equipped with these modules, SEMPO significantly reduces both pre-training data scale and model size, while achieving strong generalization. Extensive experiments on two large-scale benchmarks covering 16 datasets demonstrate the superior performance of SEMPO in both zero-shot and few-shot forecasting scenarios compared with state-of-the-art methods. Code and data are available at https://github.com/mala-lab/SEMPO.

## 1  Introduction

Time series forecasting constitutes a fundamental analytical tool within dynamic real-world systems, underpinning a diverse array of contemporary domains, such as urban computing [1], inventory optimization [2], energy demand estimation [3], and climate system modeling [4]. Forecasting has often been approached in a task-specific and end-to-end fashion. A wide range of methods have been developed in this paradigm, from classic statistical models, such as Exponential Smoothing [5] and Gaussian Processes [6], to modern deep learning approaches, including Transformer-based architectures [7–11] and those based on Multilayer Perceptrons (MLPs) [12–15]. One key issue shared by these approaches is that their effectiveness often relies heavily on large in-distribution training data and extensive domain expertise.

To mitigate this issue, a profound paradigm shift has been recently ushered in by the emergence of *foundation models* (FMs) for general-purpose forecasting, which leverage large-scale pre-training

---

\*The work was done when Hui He visited Singapore Management University
†Corresponding author

39th Conference on Neural Information Processing Systems (NeurIPS 2025).

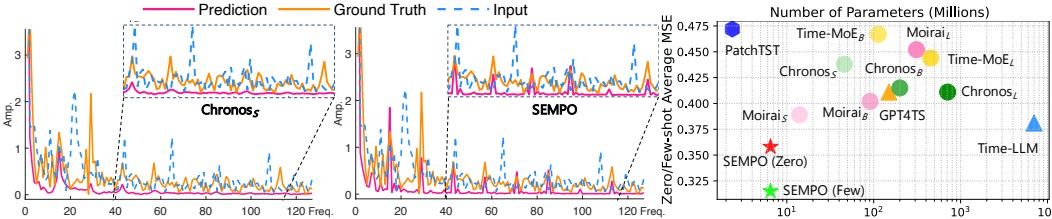

Figure 1: **Left:** Most time series FMs cannot effectively utilize low-energy frequency signals, exemplified by Chronos$_S$ [20] on the ETTh1 dataset [24]. **Middle:** The EASD module in SEMPO helps mitigate this issue. **Right:** The number of parameters in SEMPO and state-of-the-art (SOTA) time series FMs against average zero-/few-shot performance on the TSLib benchmark [25].

on massive amounts of time series data to enable *zero- and few-shot* generalization across a broad spectrum of downstream tasks. In line with this trend, research efforts have increasingly emphasized scaling up pre-training datasets and model sizes to push the frontier of transferability in diverse time series domains [16–21]. However, excessively large pre-trained models often impose substantial computational burdens during both training and inference, hindering its application to computational resource-constrained environments [22, 23]. This motivates the question: *Can we instead devise "small" FMs to serve as an alternative to these "large" FMs for zero- and few-shot forecasting in such resource-constrained environments?* A few studies [22, 23] have explored this question very recently. They showed remarkable success in reducing the model size while maintaining its generalization, but they still rely on computationally costly pre-training on large-scale data. We take this question a step further: *Can we substantially reduce both model size and pre-training data size while maintaining strong generalization ability?*

To answer this question, we propose **SEMPO**, a novel lightweight foundation model that requires pretraining on relatively small-scale data, yet exhibits strong general time series forecasting. Two key challenges on the way to this goal include **1)** significantly improved utilization of pre-training data and **2)** lightweight model architecture with strong generalizability. Two novel modules—energy-aware spectral decomposition (**EASD**) and mixture-of-prompts enabled Transformer architectures (**MoPFormer**)—are introduced in SEMPO to respectively address each of these two challenges.

Specifically, modeling full energy spectrum of temporal signals from pre-training data is essential for learning a comprehensive knowledge base of temporal patterns to underpin strong transferability across different datasets [26–28]. However, current time series FMs [17, 18, 20, 21] often suffer from bottlenecks in utilizing these pre-training data, since their pre-training can ignore low-energy frequency signals characterized by small amplitudes and distributed across mid-, high-, or even low-frequency bands (see Figure 1 **Left**). Despite being subtle, such signals are persistent and can encode stable temporal dynamics that are crucial for forecasting generalization. In light of this, we propose the **EASD** module that transforms each time series into the frequency domain and decomposes it along energy and frequency axes. To prevent low-energy components from being overwhelmed by high-energy ones, we perform adaptive spectral partitioning using a learnable energy threshold conditioned on the input's spectral characteristics. As shown in Figure 1 **Middle**, leveraging this partitioning, we apply dual-branch spectral masking on the frequency axis, where independently parameterized multi-band masks can selectively suppress high- and low-frequency bands in each energy branch.

To accommodate a wide range of heterogeneous input patterns from different domains, most time series FM solutions [16, 19, 20, 29] opt for a large Transformer architecture with vast amounts of parameters, *e.g.*, hundreds or thousands of millions of parameters, as shown in Figure 1 **Right**. Some recent works explore model specialization (*i.e.*, model customization for specific tasks or datasets) with sparse activation at various levels—such as dataset [30], frequency [17, 18], and token [21, 31]— to alleviate the computational burdens while maintaining its generalization over the heterogeneous temporal patterns. However, the architecture of their base model remains large. To address this issue, we propose the **MoPFormer** module that learns a mixture of lightweight prompt-based experts and adaptively routes different time series tokens to these experts. This enables cost-effective learning of the heterogeneous input patterns without relying on large network architectures such as the costly mixture of experts in Transformers [21, 31] in current approaches (see Figure 1 **Right**).

In summary, this work makes the following four main contributions:

- We propose SEMPO, a novel time series FM with significantly reduced model size and pre-training scale, yet demonstrating superior generalization ability on diverse downstream forecasting tasks.

- We reveal the bias towards high-energy frequency signals in the pre-training of current solutions and introduce the EASD module to mitigate this bias, enabling significantly improved data exploitation.

- We then introduce MoPFormer, a Transformer with mixture of small learnable prompts, to learn the heterogeneous temporal patterns from diverse input datasets, offering an effective lightweight alternative to popular Transformers with large mixture-of-experts networks.

- Comprehensive experiments on two large-scale benchmarks show that SEMPO, with only 6.5M parameters pre-trained on 83M time points, surpasses SOTA time series FMs with hundreds of millions of parameters pre-trained on billions of time points, demonstrating an average reduction of 12% and 22% in forecasting errors under zero- and few-shot scenarios, respectively.

## 2 Related Work

**LLM-based Time Series FMs.** The emergence of Large Language Models (LLMs), empowered by in-context learning and emergent abilities, has catalyzed a growing body of research on transferring their powerful pre-trained knowledge to the time series domain. One line of research [32,33] leverages pre-trained LLMs as zero-shot learners by directly converting numerical time series into textual inputs. Another line of research [34–37], such as GPT4TS [34], Time-LLM [30], and S$^2$IP-LLM [36], adapts LLMs to time series forecasting through prompting or fine-tuning to bridge the modality gap. Time-LLM [30] reprograms time series using text prototypes and prepends textual prompts to provide enriched contextual information. S$^2$IP-LLM [36] aligns time series embeddings with the pre-trained semantic space of LLMs by retrieving prompts from semantic anchors. Despite their positive impact on generalization compared to task-specific models (see Appendix A), these models rely on substantial computational resources.

**Pre-trained FMs on Time Series.** Foundation models pre-trained from scratch on large-scale, multi-domain time series data have recently attracted widespread attention [16–21,29,31]. Building on this paradigm, Moment [16] and Moirai [18] adopt encoder-only architectures with masked pre-training objectives, focusing on leveraging self-supervised learning to capture intricate temporal dependencies. TimesFM [17] and Timer [19] employ decoder-only architectures under a GPT-style causal modeling paradigm for autoregressive forecasting. Chronos [20] adopts an encoder-decoder architecture that tokenizes numerical time series via scaling and quantization into a fixed vocabulary. Most of these models [16,19,20,29] rely on large Transformer architectures with numerous parameters to memorize the heterogeneous input patterns across diverse domains, which can increase both the learning complexity and the costs associated with training and inference.

**Lightweight Time Series FMs.** Recently, sparse mixture of experts (MoE) has been introduced into Transformers [21, 31] to enhance generalization across the heterogeneous temporal patterns by routing diverse time series tokens to specialized experts. Despite simultaneously alleviating the computational budget with sparse activation, the architecture of their base model remains large. With the emergence of small language models [38, 39], there is also a growing interest in small time series FMs [22,23], addressing practical computational resource and cost constraints. For example, TTM [23] leverages a lightweight mixer-style architecture and pre-trains a family of compact models for general forecasting. Chronos [20] and Moirai [18] also provide small variants to tackle such real-world settings. However, they still require substantial pre-training on large-scale time series data, significantly hindering their applications in resource-constrained environments.

## 3 Methodology

### 3.1 Problem Statement

Given a time series input $\mathbf{X} = [X_{1:L}^1, X_{1:L}^2, ..., X_{1:L}^N] \in \mathbb{R}^{N \times L}$ with the number of variates $N$ and the lookback window length $L$, where $X_{1:L}^i$ denotes the historical observations of the $i$-th variate, the task of time series forecasting is to predict the future $H$ time steps $\mathbf{Y} = [X_{L+1:L+H}^1, X_{L+1:L+H}^2, ..., X_{L+1:L+H}^N] \in \mathbb{R}^{N \times H}$. This work focuses on developing a generalist time series forecaster pre-trained on a collection of $J$ datasets from diverse domains,

$\mathcal{T}_{\text{train}} = \{\mathcal{D}_{\text{train}}^1, \mathcal{D}_{\text{train}}^2, ..., \mathcal{D}_{\text{train}}^J\}$, where each $\mathcal{D}_{\text{train}}^j = \{\mathbf{X}^j, \mathbf{Y}^j\}$ represents the $j$-th dataset. For downstream evaluation, we consider two settings, including zero- and few-shot forecasting. In the few-shot setting, the pre-trained model is further adapted using a small target dataset $\mathcal{D}_{\text{tune}}$, where $\mathcal{D}_{\text{tune}} \ll \mathcal{T}_{\text{train}}$, and evaluated on a unseen $\mathcal{D}_{\text{test}}$ from the same domain. In the zero-shot setting, the model is tested directly on $\mathcal{D}_{\text{test}}$ without any additional adaptation. Following standard zero- and few-shot settings, $\mathcal{D}_{\text{tune}}$ and $\mathcal{D}_{\text{test}}$ are drawn from entirely new domains that are not presented in $\mathcal{T}_{\text{train}}$. To support any-variate forecasting, we apply channel independence [10] to decompose multivariate inputs into univariate series.

## 3.2 Overview of SEMPO

As illustrated in Figure 2, SEMPO is built upon an encoder-decoder architecture, which comprises four key components: energy-aware spectral decomposition (EASD), patchify and project, mixture-of-prompts enabled Transformer (MoPFormer), and reconstruction and prediction heads. EASD include two steps: *energy-wise spectral partitioning* and *frequency-wise spectral masking*, while MoPFormer includes a *mixture of prompts (MoP)* and a *MoPFormer block*. The training of SEMPO follows a two-stage paradigm, as outlined in Section 3.5. Below we introduce the EASD and MoPFormer components in detail.

## 3.3 EASD: Energy-aware Spectral Decomposition

Rooting in the weighting characteristics of self-attention, the Transformer backbones widely used in time series FMs [17,18,20,21] often exhibit an energy bias [40], being prone to frequency components with high energy while overlooking those with low energy. Existing forecasting methods [22,26,27] primarily focus on the bias between high- and low-frequency components along the frequency axis. To address this issue, we introduce EASD to improve spectral modeling by mitigating energy bias and enhancing the subtle yet persistent temporal patterns. It achieves this by sequentially performing energy-wise spectral partitioning and frequency-wise spectral masking as follows.

**Energy-wise Spectral Partitioning.** Given a univariate time series $X \in \mathbb{R}^L$, after instance normalization [41], the normalized input is transformed into the frequency domain by applying Fast Fourier Transform (FFT) along the temporal dimension. The spectral energy at each frequency is defined as the squared magnitude of the corresponding Fourier coefficient [40], *i.e.*, $\text{Energy}[f] = |Z[f]|^2$, where $Z$ denotes the complex spectrum. We then perform spectral partitioning in the energy axis by adjusting a learnable energy threshold $\tau$ according to the input's spectral characteristics to delineate high- and low-energy components. This separation ensures low-energy components are not overwhelmed by high-energy components, thus enabling the extraction of frequency-localized patterns for modeling complex temporal dynamics. The process can be formulated as:

$$Z = \text{FFT}(\text{Norm}(X)), \quad Z_{\text{Hec}} = Z \odot (\text{Energy}(Z) > \tau), \quad Z_{\text{Lec}} = Z - Z_{\text{Hec}}, \tag{1}$$

where $\odot$ represents element-wise multiplication along the frequency dimension, $(\text{Energy}(Z) > \tau)$ yields a binary mask that identifies frequency components to be retained in $Z_{\text{Hec}}$ if their energy exceeds the threshold $\tau$, while the remaining low-energy components are assigned to $Z_{\text{Lec}}$.

**Frequency-wise Spectral Masking.** Existing FMs [16,17,42] primarily leverage random patch masking in the temporal domain to expose models to varying context lengths, enabling the learning of multi-scale temporal dependencies and improving generalization. While this strategy enhances contextual diversity, it fails to explicitly model the structured and periodic nature inherent in time series. Inspired by the frequency decomposition learning [22], we propose a *frequency-axis spectral masking* module. In both the high- and low-energy branches, high- and low-frequency bands are selectively suppressed through independently parameterized multi-band masks. Concretely, for each energy branch, we generate a set of frequency band masks $\{M_1, M_2, ..., M_{N_M}\}$ by sampling a frequency threshold $\delta_i$ and a direction indicator $d_i$ for each mask $M_i$. The threshold $\delta_i$ defines the cutoff frequency, and the binary variable $d_i$ determines the masking direction: masking frequencies less than $\delta_i$ if $d_i = 1$, and greater than $\delta_i$ if $d_i = 0$. The corresponding mask $M_i \in \{0,1\}^{L/2+1}$ is constructed as:

$$\delta_i \sim \text{Uniform}(0, \alpha), \quad d_i \sim \text{Bernoulli}(\rho), \quad i = 1, 2, ..., N_m,$$
$$M_i[j] = \mathbf{1}[d_i \cdot (f_j \leq \delta_i) + (1 - d_i) \cdot (f_j \geq \delta_i)], \quad j = 1, 2, ..., L/2 + 1, \tag{2}$$

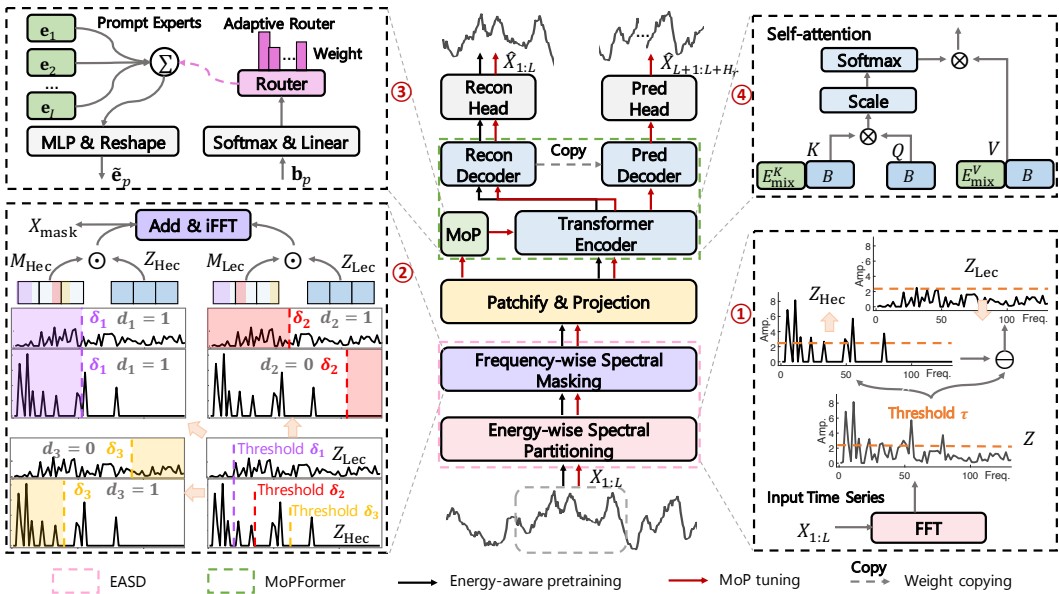

Figure 2: SEMPO follows an encoder-decoder architecture. Given a univariate time series, ① its EASD module transforms it into the frequency domain and partitions into high- and low-energy components. ② These components are independently masked across multiple frequencies, fused, and transformed back to the time domain. After patching and projection, the masked representation is fed to backbone layers and a reconstruction head as part of the energy-aware pre-training. During MoP tuning, all previous modules are frozen; only the MoP and the prediction head are trainable. The masked input is then ③ routed to token-dependent prompts via an adaptive router and ④ concatenated into the key and value of self-attention, enabling prompt-guided reconstruction and prediction.

where $\alpha < L/2 + 1$, and $f_j$ denotes the $j$-th frequency point. Stacking all $N_m$ such masks for each energy branch yields two multi-band masking matrices: $M_{\text{Hec}}, M_{\text{Lec}} \in \{0,1\}^{N_m \times (L/2+1)}$. Each row represents a distinct frequency band mask, and each element $m_{ij} \in \{0,1\}$ indicates whether the $j$-th frequency point is masked under the $i$-th mask. Correspondingly, we expand $Z_{\text{Hec}}$ and $Z_{\text{Lec}}$ along a new dimension and replicate them $N_m$ times, resulting in $Z_{\text{Hec}}, Z_{\text{Lec}} \in \mathbb{C}^{N_m \times (L/2+1)}$ that are aligned with the masking matrices. We then apply element-wise multiplication between each replicated representation and its corresponding mask, and convert the aggregated result back to the time domain using the inverse Fast Fourier Transform (iFFT):

$$X_{\text{mask}} = \text{iFFT}(Z_{\text{Hec}} \odot M_{\text{Hec}} + Z_{\text{Lec}} \odot M_{\text{Lec}}). \tag{3}$$

This masking strategy incorporates two core insights. First, $M_{\text{Hec}}$ and $M_{\text{Lec}}$ are separately generated without shared sampling parameters between the high- and low-energy branches. Such a decoupled design promotes spectral diversity and inherently mitigates energy bias. Second, by adaptively learning the sampling parameters, the masking process can be tailored to each specific time series dataset, thereby enhancing the model's ability to generalize across diverse data scenarios.

Then a standard patchify and projection step is applied. As illustrated in Figure 2, after obtaining the masked series $X_{\text{mask}} \in \mathbb{R}^{N_m \times L}$, we divide it into $P$ non-overlapping patches of equal length $L_p$, where $P = \lfloor L/L_p \rfloor$. Each patch is then projected into a patch token by a linear layer $\mathbb{R}^{L_p} \to \mathbb{R}^{D_p}$, where $D_p$ is the dimension of Transformers. The positional embedding for the $p$-th patch $\mathbf{x}_p$ is denoted by $\text{pe}_p$. Accordingly, the position-encoded patch token $\mathbf{b}_p$ is given by $\mathbf{b}_p = \text{Dropout}(\mathbf{x}_p + \text{pe}_p)$ and $B = \{\mathbf{b}_1, \mathbf{b}_2, ..., \mathbf{b}_P\} \in \mathbb{R}^{N_m \times P \times D_p}$.

## 3.4 MoPFormer: Mixture-of-prompts Enabled Transformer

A significant challenge in pre-training on multi-domain time series datasets lies in their high degree of cross-domain heterogeneity, including differences in sampling resolution, noise levels, variable semantics, etc. To overcome this limitation, we introduce *Mixture of Prompts* (MoP) and incorporate

it into stacked Transformer blocks, which learns dataset-specific prompts to guide the model's adaptation to new data, thereby promoting generalization across diverse target domains.

**Mixture of Prompts.**    To account for the diversity of multi-domain time series data, we randomly initialize a prompt-based expert pool $E = \{\mathbf{e}_1, \mathbf{e}_2, ..., \mathbf{e}_I\} \in \mathbb{R}^{I \times D_p}$, where each prompt-based expert is $\mathbf{e}_i \in \mathbb{R}^{D_p}$ with dimension $D_p$. Applying all prompt-based experts indiscriminately to each patch token may dilute the effectiveness of prompt information, particularly under significant cross-domain distributional shifts in time series data. Inspired by the conditional computation mechanism in MoE [21, 31], we introduce a token-dependent adaptive router that dynamically assigns relevant prompt-based experts conditioned on the input token. Concretely, the router computes expert-specific gating scores via a linear-softmax transformation over the input token, and performs soft merging of expert vectors through a weighted combination guided by these scores. To encode a shared structure between prompt key and value matrices [43], we further adopt a reparameterization strategy consisting of a two-layer MLP followed by a reshape operation, transforming the aggregated expert representations into structured key-value pairs. Given the masked patch token $\mathbf{b}_p \in \mathbb{R}^{D_p}$ and expert $\mathbf{e}_i \in \mathbb{R}^{D_p}$ for $i = 1, 2, ..., I$, the complete routing and merging process can be formalized as:

$$\mathbf{s}_{i,p} = \text{Softmax}(\text{Linear}(\mathbf{b}_p)), \quad \tilde{\mathbf{e}}_p = \text{Reshape}(\text{MLP}(\sum_{i=1}^{I} \mathbf{s}_{i,p} \cdot \mathbf{e}_i)), \tag{4}$$

where $\mathbf{s}_{i,p} \in \mathbb{R}^I$ denotes the gating scores over the expert pool, indicating the contribution of the $i$-th expert $\mathbf{e}_i$ to the $p$-th input token $\mathbf{b}_p$. The mixed prompt $\tilde{\mathbf{e}}_p \in \mathbb{R}^{S \times 2 \times D_p}$ represents a structured prompt key-value pairs of $\mathbf{b}_p$ distributed across $S$ Transformer layers.

**MoPFormer Block.**    Since self-attention (SA) can be interpreted as a specialized MoE [44], the mixture of prompts is integrated into the SA mechanism to inject new experts, which can be achieved via the modification of key and value matrices. Specifically, the mixed prompt $E_{\text{mix}} = \{\tilde{\mathbf{e}}_1, \tilde{\mathbf{e}}_2, ..., \tilde{\mathbf{e}}_P\} \in \mathbb{R}^{S \times 2 \times N_m \times P \times D_p}$ is first decomposed into $E_{\text{mix}}^K \in \mathbb{R}^{N_m \times P \times D_p}$ and $E_{\text{mix}}^V \in \mathbb{R}^{N_m \times P \times D_p}$. In the SA layer, let $Q$, $K$, $V$ denote the query, key, and value matrices respectively, the key and value matrices are then constructed by appending the prompt $E_{\text{mix}}^K$ and $E_{\text{mix}}^V$ to the input series $B \in \mathbb{R}^{N_m \times P \times D_p}$ along the patch dimension $P$ as follows:

$$\text{SA} = \text{Attention}(Q = B, K = \text{Concat}(E_{\text{mix}}^K, B), V = \text{Concat}(E_{\text{mix}}^V, B)). \tag{5}$$

We employ Transformer encoder [45] in both the encoder and decoder. Following SOTA time series FMs [18, 21, 31], we integrate recent advancements including RMSNorm [46], pre-normalization scheme [47], and SwiGLU [48] into our architecture. Each block at the $s$-th layer is defined as:

$$U^s = \text{SA}(\text{RMSNorm}(B^{s-1})) + B^{s-1}, \quad \bar{U}^s = \text{RMSNorm}(U^s), \quad B^s = \text{FFN}(\bar{U}^s) + U^s. \tag{6}$$

Both the encoder and decoder are constructed by stacking $S$ MoPFormer blocks. SEMPO is then pre-trained using the above steps via reconstruction and prediction objectives. More details on the MoPFormer block during pre-training and inference are provided in Appendix B.

### 3.5   Model Training

To improve the model's generalization ability in downstream zero-shot and few-shot forecasting tasks, we adopt a two-stage training paradigm: energy-aware pre-training and MoP tuning, as discussed in the following:

**Energy-aware Pre-training.**    During energy-aware pre-training, SEMPO is pre-trained on multi-domain time series data via a self-supervised reconstruction objective, enabling it to capture transferable cross-domain commonalities. As shown in Figure 2 (**black arrows**), the series representation is fed into the reconstruction decoder without incorporating the mixture of prompts. The decoder generates the reconstructed series $\hat{X} \in \mathbb{R}^L$ through a linear reconstruction head. The training objective is the Mean Squared Error (MSE) between the input and its reconstruction:

$$\mathcal{L}_{pretraining} = ||X_{1:L} - \hat{X}_{1:L}||_2^2. \tag{7}$$

Table 1: Zero-shot results on the TSLib benchmark. MSE and MAE are averaged errors over forecasting horizons $H \in \{96, 192, 336, 720\}$. '-' denotes dataset used in pre-training and excluded. Values in ( , ) denote model size and pre-training data size respectively. '$S$': Small, '$B$': Base, '$L$': Large. **Red**: the best, **Blue**: the second best. Full table is in Appendix D.

| Models | SEMPO (6.5M,83M) | | Time-MoE$_B$ (113M,309B) | | Time-MoE$_L$ (453M,309B) | | Timer (67.4M,28B) | | Moirai$_S$ (14M,27B) | | Moirai$_B$ (91M,27B) | | Moirai$_L$ (311M,27B) | | Chronos$_S$ (46M,84B) | | Chronos$_B$ (200M,84B) | | Chronos$_L$ (710M,84B) | | TimesFM (200M,100B) | | Moment (385M,1.13B) | |
|---|---|---|---|---|---|---|---|---|---|---|---|---|---|---|---|---|---|---|---|---|---|---|---|---|
| Metrics | MSE | MAE | MSE | MAE | MSE | MAE | MSE | MAE | MSE | MAE | MSE | MAE | MSE | MAE | MSE | MAE | MSE | MAE | MSE | MAE | MSE | MAE | MSE | MAE |
| ETTh1 | 0.410 | 0.430 | 0.445 | 0.449 | 0.435 | 0.449 | 0.451 | 0.463 | 0.448 | 0.432 | 0.433 | 0.431 | 0.466 | 0.443 | 0.551 | 0.463 | 0.524 | 0.439 | 0.541 | 0.443 | 0.489 | 0.444 | 0.708 | 0.580 |
| ETTh2 | 0.341 | 0.391 | 0.566 | 0.479 | 0.477 | 0.452 | 0.366 | 0.408 | 0.355 | 0.401 | 0.360 | 0.399 | 0.382 | 0.397 | 0.394 | 0.409 | 0.392 | 0.401 | 0.385 | 0.400 | 0.396 | 0.405 | 0.392 | 0.430 |
| ETTm1 | 0.503 | 0.466 | 0.507 | 0.480 | 0.483 | 0.471 | 0.544 | 0.476 | 0.554 | 0.477 | 0.566 | 0.464 | 0.601 | 0.468 | 0.628 | 0.487 | 0.566 | 0.465 | 0.521 | 0.448 | 0.434 | 0.419 | 0.697 | 0.555 |
| ETTm2 | 0.286 | 0.341 | 0.538 | 0.463 | 0.509 | 0.452 | 0.298 | 0.346 | 0.323 | 0.351 | 0.339 | 0.356 | 0.334 | 0.352 | 0.320 | 0.355 | 0.308 | 0.344 | 0.315 | 0.350 | 0.320 | 0.353 | 0.319 | 0.360 |
| Weather | 0.248 | 0.287 | 0.279 | 0.309 | 0.318 | 0.334 | 0.292 | 0.312 | 0.267 | 0.306 | 0.312 | 0.295 | 0.477 | 0.289 | 0.298 | 0.302 | 0.283 | 0.295 | 0.292 | 0.297 | - | - | 0.291 | 0.323 |
| Electricity | 0.196 | 0.295 | - | - | - | - | 0.297 | 0.375 | 0.243 | 0.329 | 0.207 | 0.296 | 0.224 | 0.309 | 0.246 | 0.312 | 0.336 | 0.329 | 0.326 | 0.328 | - | - | 0.861 | 0.766 |
| Traffic | 0.466 | 0.344 | - | - | - | - | 0.613 | 0.407 | - | - | - | - | - | - | 0.614 | 0.420 | 0.603 | 0.413 | 0.600 | 0.411 | - | - | 1.411 | 0.804 |
| $1^{st}$ Count | 12 | | 0 | | | | 0 | | | | 0 | | | | | | 0 | | | | 2 | | 0 | |

**MoP Tuning.** During MoP tuning, SEMPO is jointly trained with supervised forecasting and self-supervised reconstruction to adapt to domain-specific variations by tuning both the mixture of prompts and the prediction heads, while keeping the Transformer backbone frozen, as shown in Figure 2 (**red arrows**). We adopt a multi-resolution forecasting strategy [21] with each prediction head implemented as a single linear layer. A composite loss, aggregating forecasting errors over different horizons, is computed to enhance model generalization. Let $\mathcal{H} = \{H_1, H_2, ..., H_R\}$ denote the set of forecasting horizons, where $H_r$ is the number of future steps for the $r$-th resolution. Given the ground truth $\hat{X}_{L+1:L+H_r}$ for $r = 1, 2, ..., R$, the training loss is then defined as:

$$\mathcal{L}_{tuning} = \sum_{H_r \in \mathcal{H}} ||X_{L+1:L+H_r} - \hat{X}_{L+1:L+H_r}||_2^2 + ||X_{1:L} - \hat{X}_{1:L}||_2^2. \tag{8}$$

# 4 Experiments

**Datasets.** Leveraging large-scale publicly available time series collection UTSD [19], we curate a diverse subset covering multiple domains, totaling ∼83 million time points. We then set the pre-training training-validation split to 9:1, following [19]. For zero-/few-shot forecasting, we evaluate SEMPO on the Time-Series-Library (TSLib) benchmark [25], which includes seven datasets: ETTh1, ETTh2, ETTm1, ETTm2, Weather, Electricity, and Traffic. Given that most real-world scenarios involve limited data, we draw on prior works [16, 17, 20, 22] and set the lookback window length $L = 512$ on TSLib. We also include GIFT-Eval [49], the largest benchmark for zero-shot forecasting [50]. To avoid data leakage, we remove datasets overlapping with pre-training corpus and TSLib, and select 9 datasets from the GIFT-Eval benchmark spanning diverse domains. For GIFT-Eval, we adhere to its default evaluation protocol [49]. More details are given in Appendix C.1.

**Baselines.** We evaluate SEMPO against 17 latest open-sourced SOTA forecasting models, grouped in three categorized: (1) **Pre-trained FMs on time series**: TTM [23], Time-MoE [21], Timer [19], Moirai [18], Chronos [20], TimesFM [17], and Moment [16]; (2) **LLM-based time series FMs**: Time-LLM [30], GPT4TS [34], and S$^2$IP-LLM [36]; (3) **Task-specific models**: iTransformer [11], DLinear [12], PatchTST [10], TimesNet [25], Stationary [9], FEDformer [8], and Autoformer [7]. Among these, the TTM family and small variants of Moirai and Chronos are lightweight pre-trained FMs on time series. More baseline details are provided in Appendix C.2.

**Implementation Details.** Using 83M pre-training datasets, the entire two-stage pre-training process takes 10 hours on 4 A6000-48G GPUs with BF32 precision and a batch size of 2,048. By default, we set layer_number $S = 6$, head_number=16, latent_dimension $D_p = 256$, patch_size $L_p = 64$, prompt_number $I = 128$, and mask_number $N_M = 4$. The hyperparameter analysis of $I$, $N_M$, and $D_p$, as well as the scaling analysis of the model are given in Appendix D. Regarding optimization, we use AdamW optimizer with hyperparameters: learning_rate=1e-3, weight_decay=0.1, $\beta_1 = 0.9$, $\beta_2 = 0.95$. A constant learning rate is used after a linear warmup for the first 10,000 steps. More implementation details are provided in Appendix C.3.

Table 2: Few-shot results on the TSLib benchmark with 5% training data. MSE and MAE are averaged over forecasting horizons $H \in \{96, 192, 336, 720\}$, where lower values indicate better prediction. **Red**: the best, **Blue**: the second best. Full table is in Appendix D.

| Models | SEMPO | | TTM | | Time-LLM | | GPT4TS | | S²IP-LLM | | iTransformer | | DLinear | | PatchTST | | TimesNet | | Stationary | | FEDformer | | Autoformer | |
|---|---|---|---|---|---|---|---|---|---|---|---|---|---|---|---|---|---|---|---|---|---|---|---|---|
| Metrics | MSE | MAE | MSE | MAE | MSE | MAE | MSE | MAE | MSE | MAE | MSE | MAE | MSE | MAE | MSE | MAE | MSE | MAE | MSE | MAE | MSE | MAE | MSE | MAE |
| ETTh1 | 0.406 | 0.423 | 0.382 | 0.405 | 0.627 | 0.543 | 0.681 | 0.560 | 0.642 | 0.546 | 1.070 | 0.710 | 0.750 | 0.611 | 0.694 | 0.569 | 0.925 | 0.647 | 0.943 | 0.646 | 0.658 | 0.562 | 0.722 | 0.598 |
| ETTh2 | 0.320 | 0.372 | 0.333 | 0.376 | 0.382 | 0.418 | 0.400 | 0.433 | 0.380 | 0.415 | 0.488 | 0.475 | 0.694 | 0.577 | 0.827 | 0.615 | 0.439 | 0.448 | 0.470 | 0.489 | 0.463 | 0.454 | 0.441 | 0.457 |
| ETTm1 | 0.363 | 0.385 | 0.389 | 0.389 | 0.425 | 0.434 | 0.472 | 0.450 | 0.416 | 0.421 | 0.784 | 0.597 | 0.400 | 0.417 | 0.526 | 0.476 | 0.717 | 0.561 | 0.857 | 0.598 | 0.730 | 0.592 | 0.796 | 0.620 |
| ETTm2 | 0.256 | 0.315 | 0.285 | 0.328 | 0.274 | 0.323 | 0.308 | 0.346 | 0.279 | 0.325 | 0.356 | 0.388 | 0.399 | 0.426 | 0.314 | 0.352 | 0.344 | 0.372 | 0.341 | 0.372 | 0.381 | 0.404 | 0.388 | 0.433 |
| Weather | 0.230 | 0.268 | 0.236 | 0.273 | 0.260 | 0.309 | 0.263 | 0.301 | 0.257 | 0.295 | 0.309 | 0.339 | 0.263 | 0.308 | 0.269 | 0.303 | 0.298 | 0.318 | 0.327 | 0.328 | 0.309 | 0.353 | 0.310 | 0.353 |
| Electricity | 0.165 | 0.263 | 0.183 | 0.276 | 0.179 | 0.268 | 0.178 | 0.273 | 0.186 | 0.281 | 0.201 | 0.296 | 0.176 | 0.275 | 0.181 | 0.277 | 0.402 | 0.453 | 0.627 | 0.603 | 0.266 | 0.353 | 0.346 | 0.404 |
| Traffic | 0.410 | 0.287 | 0.427 | 0.303 | 0.423 | 0.298 | 0.434 | 0.305 | 0.419 | 0.298 | 0.450 | 0.324 | 0.450 | 0.317 | 0.418 | 0.296 | 0.867 | 0.493 | 1.526 | 0.839 | 0.676 | 0.423 | 0.833 | 0.502 |
| $1^{st}$ Count | 12 | | 2 | | 0 | | 0 | | 0 | | 0 | | 0 | | 0 | | 0 | | 0 | | 0 | | 0 | |

## 4.1 Zero-shot Forecasting

**Setup.** In this section, we evaluate the performance of SEMPO for zero-shot forecasting on two representative benchmarks: TSLib [25] and GIFT-Eval [49], using MAE (Mean Absolute Error) and MSE for TSLib, and SMAPE (Symmetric Mean Absolute Percentage Error) and NRMSE (Normalized Root Mean Squared Error) for GIFT-Eval.

**Results.** Table 1 reports the average zero-shot forecasting results across multiple horizons $H \in \{96, 192, 336, 720\}$ for Transformer-based FMs. Additional results comparing the MLP-based FMs TTM and different lookback window lengths are provided in Appendix D. Despite containing only 6.5M parameters and using only 83M pre-training data, SEMPO is consistently the best performer on the majority of datasets, yielding average reductions of 23.1% in MSE and 10.8% in MAE across all methods. In particular, compared to lightweight FMs with similar parameter scale, *e.g.*, Moirai$_S$ (14M, 27B), SEMPO exhibits markedly superior performance. Remarkably, even when benchmarked against large-scale time series FMs such as Chronos$_L$ (710M, 84B), Time-MoE$_L$ (453M, 309B) and Moment (385M, 1.13B),

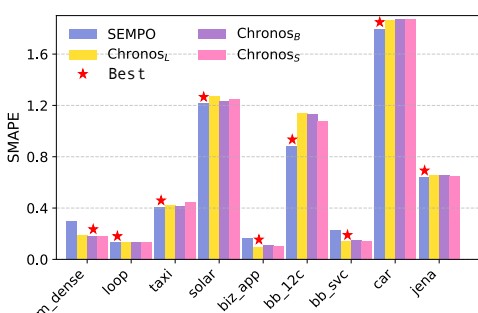

Figure 3: Zero-shot results on the GIFT-Eval benchmark. Full details in Appendix D.

SEMPO achieves substantial MSE improvements of 19.9%, 19.2%, and 36.0%, respectively; these improvements are even more pronounced when comparing the pre-training data scale: 83M of time points in SEMPO vs. 1.13B to 309B in these large FMs. Besides, Figure 3 presents a comparison between SEMPO and Chronos family on GIFT-Eval. SEMPO achieves the best performance in most cases (6 out of 9). This promising result also demonstrates that SEMPO learns a powerful knowledge base, which serves as a solid foundation for its remarkable capabilities in knowledge transfer.

## 4.2 Few-shot Forecasting

**Setup.** We fine-tune the pre-trained SEMPO and baseline models on the training splits of seven datasets from the TSLib benchmark. Following the few-shot protocols in [23, 30, 34], we consider two training scenarios using 5% and 10% of the training data, respectively. In both cases, only the prediction head and the MoP module are fine-tuned.

**Results.** Table 2 presents the average results under the 5% few-shot setting, with the corresponding 10% results provided in Appendix D. SEMPO establishes its superiority by outperforming 11 recent deep models across both 5% and 10% scenarios, achieving an average MSE reduction of 32.3%. In particular, SEMPO significantly outperforms the LLM-based pre-trained models Time-LLM (11.4%), GPT4TS (15.9%), and S²IP-LLM (10.7%), and surpasses the lightweight MLP-based FM TTM by 4.6%. These results underscore the central contribution of SEMPO's pre-trained initialization,

Table 3: Ablation studies. Zero-shot MSE and MAE averaged over $H \in \{96, 192, 336, 720\}$ on the TSLib benchmark, evaluated with different model components. Full table is in Appendix D.

| Design | ETTh1 | | ETTh2 | | ETTm1 | | ETTm2 | | Weather | | Electricity | | Traffic | |
|---|---|---|---|---|---|---|---|---|---|---|---|---|---|---|
| | MSE | MAE | MSE | MAE | MSE | MAE | MSE | MAE | MSE | MAE | MSE | MAE | MSE | MAE |
| SEMPO | 0.410 | 0.430 | 0.341 | 0.391 | 0.503 | 0.466 | 0.286 | 0.341 | 0.248 | 0.287 | 0.196 | 0.295 | 0.466 | 0.344 |
| **A.1** Multi-band Masking | 0.462 | 0.461 | 0.423 | 0.436 | 0.562 | 0.493 | 0.342 | 0.376 | 0.261 | 0.308 | 0.204 | 0.306 | 0.537 | 0.356 |
| **A.2** Random Patch Masking | 0.446 | 0.460 | 0.400 | 0.428 | 0.574 | 0.498 | 0.340 | 0.381 | 0.261 | 0.313 | 0.243 | 0.345 | 0.647 | 0.404 |
| **B.1** Sparse MoE (3 experts, 1 activated) | 0.441 | 0.452 | 0.358 | 0.402 | 0.515 | 0.485 | 0.308 | 0.360 | 0.253 | 0.292 | 0.223 | 0.321 | 0.532 | 0.391 |
| **B.2** Prefix Tuning | 0.430 | 0.448 | 0.359 | 0.404 | 0.513 | 0.475 | 0.309 | 0.361 | 0.268 | 0.306 | 0.217 | 0.315 | 0.494 | 0.365 |

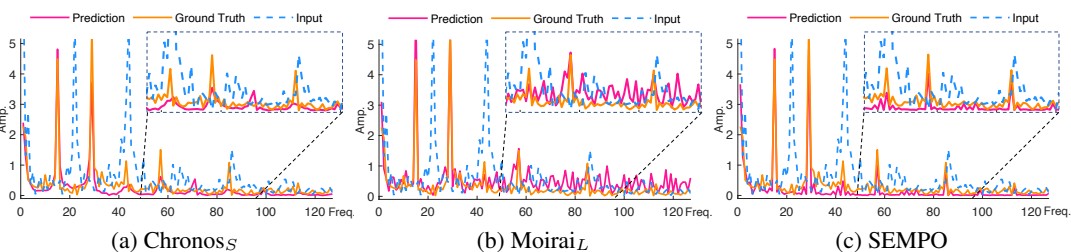

(a) Chronos$_S$        (b) Moirai$_L$        (c) SEMPO

Figure 4: Spectrum visualizations of predictions from Chronos$_S$, Moirai$_L$, and SEMPO on Electricity.

which equips the model with strong inductive biases and significantly enhances its effectiveness in resource-constrained scenarios. Compared to the zero-shot results in Table 1, fine-tuning yields only marginal gains on ETTh1 and ETTh2, suggesting that adaptation is less effective when the target dataset is limited in scale or variability.

## 4.3 Model Analysis

**Ablation Study.** To verify the design of SEMPO, we perform a series of ablation studies under the zero-shot setting, each targeting a key model module. As shown in Table 3, replacing the default dual-branch spectral masking leads to consistent performance degradation: multi-band masking [22] (A.1) and random patch masking [10] (A.2) increase the average MSE from 0.350 to 0.399 and 0.416, respectively. The degradation in A.2 stems from its unstructured nature, which disrupts temporal dependencies and corrupts spectral coherence. While A.1 retains frequency-wise modeling, the lack of energy-based partitioning allows the dominant high-energy components to eclipse subtle yet informative low-energy signals, resulting in large information loss. Furthermore, we evaluate the impact of replacing the MoP module with two variants. The first (B.1) substitutes MoP with a sparse MoE [21] with three experts and one activated, totaling 8.5M parameters. Despite the larger model size, it underperforms SEMPO, highlighting the superior efficiency and expressiveness of the MoP design. In the second variant (B.2), we remove the adaptive router in MoP and replace it with conventional prefix tuning [51], which leads to further degradation and underscores the importance of data-dependent adaptive router in enhancing generalization across diverse time series patterns.

**Analysis of Low-energy Components.** We further investigate SEMPO's capability to model low-energy components in temporal signals. Figure 4 shows the zero-shot prediction results in the spectral domain for the Electricity dataset with prediction length $H = 336$. As shown in Figure 4, Chronos$_S$ and Moirai$_L$—the small and large variants of the Chronos and Moirai families, respectively—tend to focus on dominant high-energy components. Chronos$_S$ largely neglects subtle yet persistent low-energy components. Moirai$_L$ exhibits non-negligible responses in low-energy regions, yet these do not consistently align with the ground-truth spectral peaks. This indicates that while Moirai$_L$ can partially capture low-energy signals, it still predominantly attends to high-energy components, resulting in less accurate learning of low-energy features such as important positional information. In contrast, SEMPO consistently attends to signals across the full energy spectrum, more effectively preventing low-energy components from being overwhelmed than both compact and large FMs. Besides, as shown in Appendix D, Figure 8, predictions on ETTh1 dataset across various prediction

lengths demonstrate SEMPO's strong ability to identify and leverage low-energy regions, regardless of whether they reside in high-, mid-, or even low-frequency bands of the spectrum.

**Visualization of MoP.** To analyze MoP's ability to learn dataset-specific knowledge, we visualize the soft gating scores across different datasets in Figure 5. Within each dataset, different patch tokens attend to different subsets of prompt-based experts, with each expert specializing in distinct knowledge, indicating that MoP enables fine-grained, token-level composition of prompt information. When comparing across datasets, same-domain cases (*e.g.*, ETTh1 and ETTm2) exhibit similar prompt routing patterns, while cross-domain datasets (*e.g.*, Traffic and Weather) reveal distinct token-to-prompt mappings, reflecting domain-specific adaptation. The heterogeneous routing patterns indicate that the model dynamically adapts its representations to the specific characteristics of each dataset, supporting SEMPO's strong generalization and transferability.

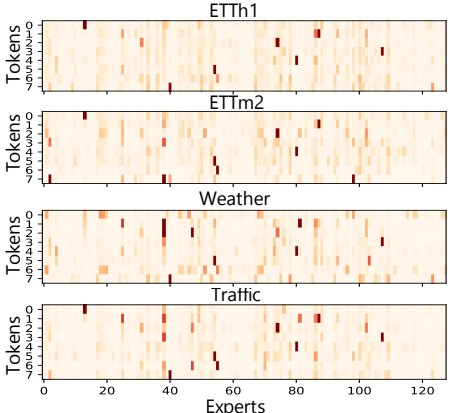

Figure 5: Gating scores of various experts.

**Efficiency Analysis.** To highlight the performance and efficiency benefits of SEMPO, we compare its inference costs with other FMs and assess their performance on ETTh1. The zero-shot results are shown in Figure 6. SEMPO, a lightweight general model with only 6.5M parameters and a fast inference time of 22s, significantly outperforms all other FMs. Compared to Moirai$_S$, the smallest model among them, SEMPO achieves nearly 10 times faster inference speed while delivering superior prediction performance. In contrast to large-scale FMs, SEMPO precisely fulfills the urgent need for general models in resource-constrained real-world scenarios, where both exceptional efficiency and superior accuracy are indispensable. More efficiency results are in Appendix D.

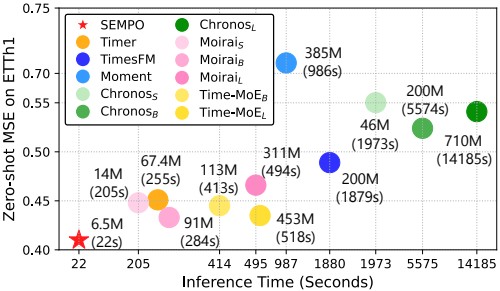

Figure 6: Efficiency comparison on ETTh1.

## 5 Conclusion

In this paper, we propose SEMPO, a novel lightweight foundation model for general time series forecasting, with relatively small data scale and model size while exhibiting strong generalization capabilities. SEMPO leverages two novel modules: energy-aware spectral decomposition for significantly improved data exploitation during pre-training, and mixture-of-prompts enabled Transformer for parameter-efficient model adaptation across different dataset and domains. Extensive experiments on two widely-used benchmarks spanning 16 datasets highlight the superiority of SEMPO in term of both effectiveness and efficiency for zero- and few-shot forecasting tasks. This work paves the way for future advancements in considering multivariate interactions and flexible distribution forecasting, which will empower the model structure with more time series capabilities.

## Acknowledgments

This research is supported by the National Natural Science Foundation of China (62272048), A*STAR under its MTC YIRG Grant (M24N8c0103), the Ministry of Education, Singapore under its Tier-1 Academic Research Fund (24-SIS-SMU-008), and the Lee Kong Chian Fellowship.

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

## A  Further Related Work

**Task-specific Deep Models for Time Series.**  In recent years, deep learning has revolutionized time series forecasting by enabling models to capture complex temporal dynamics and nonlinear dependencies [52]. Early efforts explored various neural architectures for modeling temporal patterns, such as Recurrent Neural Networks (RNNs) [1, 53, 54], Convolution Neural Networks (CNNs) [25, 55, 56], etc. Thereafter, the limited ability of these models to capture long-range dependencies has motivated a shift towards Transformer-based [7–11] and MLP-based [12–15] architectures, which have achieved SOTA performance across a wide range of forecasting tasks. While these models demonstrate competitive in-distribution performance, they typically require meticulous hyperparameter tuning across different datasets and exhibit limited flexibility and generalizability when applied to out-of-distribution tasks, particularly under few-shot or zero-shot scenarios. Time series foundation models help address this challenge.

## B  More Details about MoPFormer Block

Recent insights from vision research [44] reveal that self-attention (SA) can be interpreted as a specialized MoE architecture, characterized by linear experts and quadratic gating score functions. Building on this connection, we propose that applying a mixture of prompts (MoP)—in a manner similar to prefix-tuning [51]—within pre-trained Transformer backbones serves as a flexible and effective mechanism for injecting new experts. These newly added experts work in conjunction with the pre-trained experts, facilitating efficient adaptation of the model to new and unseen datasets.

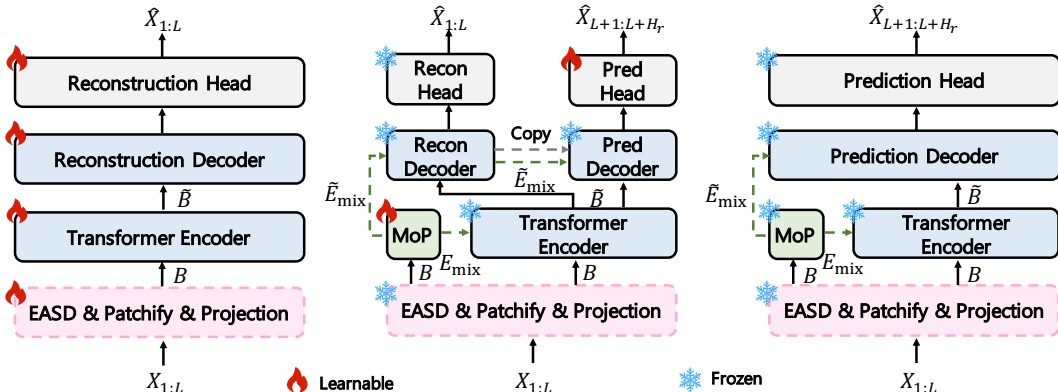

Figure 7: MoPFormer block details at different stages. **Left:** Energy-aware pre-training stage. **Middle:** MoP tuning stage. **Right:** Inference stage.

As shown in Figure 7 **Left** and **Middle** above, MoP is introduced during the MoP tuning stage. Both the encoder and decoder utilize the Transformer [45] encoder structure, as adopted in [22]. They are constructed by stacking $S$ MoPFormer blocks. Each MoPFormer block integrates recent advancements from SOTA time series foundation models [18, 21, 31]. Specifically, all LayerNorm layers are replaced with RMSNorm [46] and applied pre-normalization scheme [47] to mitigate training instability in deep architectures. The standard non-linearity in FFN layers is replaced with SwiGLU [48], which improves expressiveness and stability while preserving parameter efficiency. During the tuning stage, given a univariate time series $X \in \mathbb{R}^L$, instance normalization is first applied with zero mean and unit standard deviation to mitigate distribution shifts [41]. Recent work [40] has theoretically shown that normalization scales the absolute spectral energy by the variance but does not affect its relative distribution. Then, after the proposed EASD module and the Patchify & Projection module, the encoder processes the masked series representation $B \in \mathbb{R}^{N_m \times P \times D_p}$ along with the mixed prompt $E_{\mathrm{mix}} \in \mathbb{R}^{S \times 2 \times N_m \times P \times D_p}$. The decoder first aggregates information over the $N_m$ dimension via an *averaging operator* and then takes the aggregated series $\tilde{B} \in \mathbb{R}^{P \times D_p}$, along with its corresponding mixed prompt $\tilde{E}_{\mathrm{mix}} \in \mathbb{R}^{S \times 2 \times P \times D_p}$ as input.

In downstream fine-tuning, the model is trained on the target dataset using the same joint training strategy as in the MoP tuning stage, in a few-shot setting. For zero-shot inference, the reconstruction

decoder and head are removed, and all modules are frozen. A greedy scheduling algorithm [21] is then applied to generate predictions for arbitrary target lengths, as shown in Figure 7 **Right**.

## C  Experimental Details

### C.1  Datasets

**Pretraining Datasets.**   Large-scale publicly available time series collections, *e.g.*, Time-300B [21], UTSD [19], and LOTSA [18], have resolved data availability issues for building general time series forecasting models. Leveraging UTSD [19], we curate a diverse subset covering multiple domains, totaling ~83 million time points. We then set the pre-training training-validation split to 9:1, following [19]. Below, we outline the key properties of the datasets, including their domain, resolution, number of time points, total file size, Augmented Dickey-Fuller (ADF) test statistics, forecastability, and data source.

Table 4: List of pre-training datasets. All datasets are drawn from a subset of the UTSD [19] data repositories, and all attribute specifications (*e.g.*, resolution, time points, file size, ADF., forecast) adhere to the original UTSD schema.

| Datatset | Domain | Resolution | Time Points | File Size | ADF. | Forecast. | Source |
|---|---|---|---|---|---|---|---|
| Australian Electricity Demand | Energy | 30 Min | 1.16M | 5M | -27.554 | 0.730 | Monash [57] |
| Wind | Energy | 4 Sec | 7.40M | 29M | -29.174 | 0.811 | Monash [57] |
| KDD Cup 2018 | Nature | Hourly | 2.94M | 12M | -10.107 | 0.362 | Monash [57] |
| Temperature Rain | Nature | Daily | 23.25M | 93M | -10.952 | 0.133 | Monash [57] |
| Saugeen River Flow | Nature | Daily | 0.02M | 1M | -19.305 | 0.300 | Monash [57] |
| Sunspot | Nature | Daily | 0.07M | 1M | -7.866 | 0.287 | Monash [57] |
| PIGCVP | Health | - | 0.62M | 3M | -4.855 | 0.577 | UCR [58] |
| US Births | Health | Daily | 0.00M | 1M | -3.352 | 0.675 | Monash [57] |
| PEMS04 | Transport | 5 Min | 15.65M | 60M | -15.192 | 0.494 | [56] |
| PEMS07 | Transport | 5 Min | 24.92M | 96M | -20.603 | 0.466 | [56] |
| Pedestrian Counts | Transport | Hourly | 3.13M | 12M | -23.462 | 0.297 | Monash [57] |
| Beijing PM25 Quality | Environment | Hourly | 3.66M | 14M | -31.415 | 0.404 | [59] |

**Evaluation Datasets.**   For zero- and few-shot forecasting, we evaluate SEMPO on the widely-used Time-Series-Library (TSLib) benchmark [25], which includes seven datasets: *ETTh1, ETTh2, ETTm1, ETTm2, Weather, Electricity, and Traffic*. In detail, the ETT[3] (Electricity Transformer Temperature) dataset is collected from electric power industry and contains 2 years of data from two separate counties in China. It includes temperature measurements and power load features. The dataset is further divided into subsets for different time granularities: {ETTh1, ETTh2} for 1-hour intervals and {ETTm1, ETTm2} for 15-minute intervals. The Weather[4] dataset contains 21 meteorological indicators such as air temperature and humidity, is collected every 10 minutes from the Weather Station of the Max Planck Institute for Biogeochemistry in 2020. The Electricity[5] dataset includes hourly electricity consumption data from 321 clients, measured in kWh. Due to missing data, the dataset was adjusted to an hourly granularity over two years. The Traffic[6] dataset collects 48 months (2015-2016) of hourly data from the California Department of Transportation, which tracks road occupancy rates on San Francisco Bay Area freeways. The details about these datasets are summarized in Table 5.

We also include GIFT-Eval  [49] for evaluation. This comprehensive benchmark is designed to evaluate general-purpose time series forecasting models, particularly foundation models. It comprises 23 evaluation datasets, comprising approximately 144,000 time series and 177 million data points, spanning seven domains—economics/finance, energy, healthcare, nature, sales, transportation, and Web/CloudOps—and covers ten sampling frequencies, ranging from seconds to yearly. The benchmark supports both univariate (15 datasets) and multivariate (8 datasets) forecasting tasks with *short-*, *medium-*, and *long-term* prediction lengths. GIFT-Eval also provides a rich set of statistical

---

[3]https://github.com/zhouhaoyi/ETDataset
[4]https://www.bgc-jena.mpg.de/wetter/
[5]https://archive.ics.uci.edu/ml/datasets/ElectricityLoadDiagrams20112014
[6]http://pems.dot.ca.gov

Table 5: List of datasets TSLib benchmark.

| Dataset | ETTh1 | ETTh2 | ETTm1 | ETTm2 | Weather | Electricity | Traffic |
|---|---|---|---|---|---|---|---|
| Variables | 7 | 7 | 7 | 7 | 21 | 321 | 862 |
| Time Points | 17420 | 17420 | 69680 | 69680 | 52696 | 26304 | 17544 |
| Resolution | Hourly | Hourly | 15 Min | 15 Min | 10 Min | Hourly | Hourly |
| Domain | Electricity | Electricity | Electricity | Electricity | Weather | Electricity | Traffic |

features (*e.g.*, trend, seasonality, entropy, stability) for in-depth data analysis. To avoid data leakage, we remove datasets overlapping with pre-training corpus and TSLib, and select 9 datasets including *m_dense, loop_seattle, sz_taxi, solar, bizitobs_application, bizitobs_12c, bitbrains_service, car_parts, and jena_weather*. For GIFT-Eval, we adhere to its default evaluation protocol [49].

## C.2 Baselines

**Pre-trained FMs on time series.** The models include TTM [23], Time-MoE [21], Timer [19], Moirai [18], Chronos [20], TimesFM [17], and Moment [16], detailed as follows. A comparison between these pre-trained FMs is presented in Table 6. Notably, although SEMPO is evaluated with fixed context lengths in this paper—similar to TTM [23]—its Transformer-based architecture is inherently insensitive to context length variations, as it can accommodate different context lengths through preprocessing techniques such as padding [20].

**TTM** [23] is a MLP-based compact model tailored for zero- and few-shot multivariate time series forecasting. The TTM family includes three variants: $TTM_B$ (1M), $TTM_E$ (4M), and $TTM_A$ (5M). The official checkpoint is available at `https://huggingface.co/ibm-granite/granite-timeseries-ttm-r2`.

**Time-MoE** [21] introduces a MoE structure into the Transformer framework, enabling sparse activation where only a subset of experts is activated per input token. The Time-MoE family includes three variants: Time-MoE$_{base}$ (113M), Time-MoE$_{large}$ (453M), and Time-MoE$_{ultra}$ (2.4B). The official checkpoint is available at `https://huggingface.co/Maple728`.

**Timer** [19] adopts the decoder-only Transformer architecture and is pre-trained with a next token prediction objective on massive time series corpora. The official checkpoint is available at `https://huggingface.co/collections/thuml/time-series-foundation-models-67c80ace73299239b651d954`.

**Moirai** [18] is a universal forecasting model designed to handle diverse time series forecasting tasks in a zero-shot manner. The Moirai family includes three variants: Moirai$_{small}$ (14M), Moirai$_{base}$ (91M), and Moirai$_{large}$ (311M). The official checkpoint is available at `https://huggingface.co/collections/Salesforce/moirai-r-models-65c8d3a94c51428c300e0742`.

**Chronos** [20] is designed for zero-shot and in-domain forecasting, and demonstrates that by learning the 'language' of time series. The Chronos family includes three variants: Chronos$_{small}$ (46M), Chronos$_{base}$ (200M), and Chronos$_{large}$ (710M). The official checkpoint is available at `https://huggingface.co/collections/amazon/chronos-models-and-datasets-65f1791d630a8d57cb718444`.

**TimesFM** [17] introduces input patching to represent sequences efficiently, leverages longer output patches to reduce autoregressive steps, and employs random masking to learn from variable-length contexts. The official checkpoint is available at `https://huggingface.co/google/timesfm-1.0-200m`.

**Moment** [16] leverages a patch-based Transformer encoder trained with masked time series modeling to learn robust representations across diverse domains. The Moment family includes three variants: Moment$_{small}$ (40M), Moment$_{base}$ (125M), and Moment$_{large}$ (385M). The official checkpoint is available at `https://huggingface.co/AutonLab`.

**LLM-based time series FMs.** The models include Time-LLM [30], GPT4TS [34], and $S^2$IP-LLM [36], detailed as follows:

Table 6: Comparison between pre-trained FMs on time series.

| Method | SEMPO | TTM | Time-MoE | Timer | Moirai | Chronos | TimesFM | Moment |
|---|---|---|---|---|---|---|---|---|
| Architecture | Encoder-only | Mixer-style | Decoder-only | Decoder-only | Encoder-only | Encoder-decoder | Decoder-only | Encoder-only |
| (Max) Model Size | 9.9M | 5M | 2.4B | 67M | 311M | 710M | 200M | 385M |
| (Min) Model Size | 6.5M | 1M | 113M | 67M | 14M | 46M | 200M | 40M |
| Input Token | Patch | Patch | Point | Patch | Patch | Point | Patch | Patch |
| Dataset Scale | 83M | 1B | 309B | 28B | 27B/231B | 84B | 100B | 1.13B |
| Context Length | 512/1024/1536 | 512/1024/1536 | $\leq$4096 | $\leq$1440 | $\leq$5000 | $\leq$512 | $\leq$512 | 512 |
| Source | Ours | [23] | [21] | [19] | [18] | [20] | [17] | [16] |

**Time-LLM** [30] is a reprogramming framework that adapts off-the-shelf LLMs to perform time series forecasting. Time-LLM (LLaMA full, 3.4B), Time-LLM (LLaMA-8, 976M), Time-LLM (GPT-2, 124M). The official implementation is available at `https://github.com/KimMeen/Time-LLM`.

**GPT4TS** [34] fine-tunes the input embedding, positional embeddings, layer normalization, and output layer to adapt to time series data. It has about 4M learnable parameters (total 87M). The official implementation is available at `https://github.com/DAMO-DI-ML/One_Fits_All`.

**S$^2$IP-LLM** [36] aims to enhance time series forecasting by aligning time series embeddings with the semantic space of a pre-trained large language model. The total parameters are 117M. The official implementation is available at `https://github.com/panzijie825/S2IP-LLM`.

**Task-specific Models.** The models include iTransformer [11], DLinear [12], PatchTST [10], TimesNet [25], Stationary [9], FEDformer [8], and Autoformer [7], detailed as follows:

**iTransformer** [11] inverts the conventional design by embedding the entire time series of each variate as a token, enabling attention mechanisms to directly model multivariate correlations. The official implementation is available at `https://github.com/thuml/iTransformer`.

**DLinear** [12] directly maps the historical input sequence to future outputs via a temporal linear projection, without modeling inter-variable dependencies. The official implementation is available at `https://github.com/cure-lab/LTSF-Linear`.

**PatchTST** [10] segments each univariate series into subseries-level patches and allows each time series channel to be processed separately using a shared Transformer backbone. The official implementation is available at `https://github.com/yuqinie98/PatchTST`.

**TimesNet** [25] introduces temporal 2D-variation modeling by transforming 1D time series into structured 2D tensors based on automatically discovered periods. The official implementation is available at `https://github.com/thuml/TimesNet`.

**Stationary** [9] integrates a series stationarization module to normalize input sequences and the de-stationary attention mechanism by re-incorporating statistical properties into the attention computation. The official implementation is available at `https://github.com/thuml/Nonstationary_Transformers`.

**FEDformer** [8] combines seasonal-trend decomposition with frequency-domain modeling to enhance the global understanding of temporal dynamics. The official implementation is available at `https://github.com/MAZiqing/FEDformer`.

**Autoformer** [7] replaces point-wise self-attention with a series-wise auto-correlation operation that captures periodic dependencies across sub-series using fast Fourier transform. The official implementation is available at `https://github.com/thuml/Autoformer`.

## C.3 Implementation Details

SEMPO and all other baselines are conducted on $4\times$ NVIDIA A6000-48G GPUs. The feedforward dimension d_ff is set to 256, and head dropout is set to 0.2. During energy-aware pre-training, the model is trained for 10 epochs, with a batch size of 2,048. MoP tuning is performed for 20 epochs, with the same batch size of 2,048. For few-shot and zero-shot settings, the batch size is reduced to 32. Early stopping is applied with a patience of 6. All experimental results are obtained by averaging

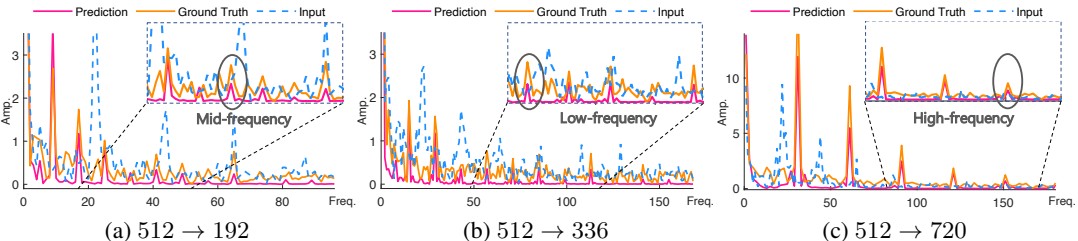

Figure 8: Spectrum visualizations of predictions on ETTh1 with different prediction lengths.

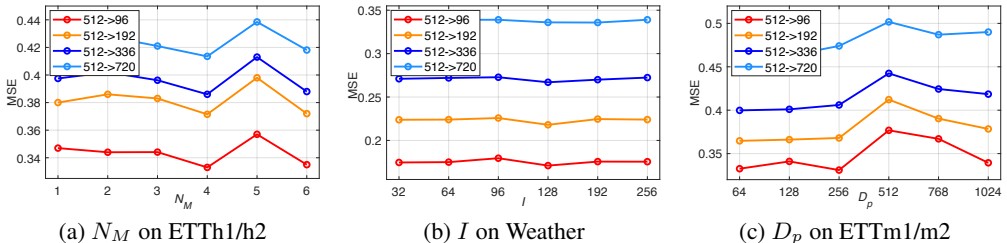

Figure 9: Sensitivity analysis of mask_number $N_M$, prompt_number $I$, and latent_dimension $D_p$.

over three runs with different random seeds. In deep time series forecasting, performance variance across seeds is typically small, and following prior work [19, 21, 23], we report only the mean values.

## D  Additional Results

### D.1  Analysis of Low-energy Components

We further investigate SEMPO's capability to model low-energy components in temporal signals across diverse prediction lengths. Experiments are conducted on the ETTh1 dataset, with the results presented in Figure 8. These figures demonstrate that SEMPO exhibits full energy spectrum learning capability, applicable to both short- and long-term forecasting. Notably, SEMPO effectively identifies and leverages low-energy regions, regardless of whether they reside in high-, mid-, or even low-frequency bands of the spectrum.

### D.2  Sensitivity Analysis

The parameters mask_number ($N_M$ in Equation 2), prompt_number ($I$ in Equation 4), and latent_dimension ($D_p$ in Patchify & Projection module) play a pivotal role in shaping the functionality of EASD and MoPFormer. In this section, we conduct comprehensive experiments on the ETTh1/h2, ETTm1/m2, and Weather datasets to investigate the impact of these parameters on forecasting performance. We explore a range of values from the sets {1, 2, 3, 4, 5, 6}, {32, 64, 96, 128, 192, 256}, and {64, 128, 256, 512, 768, 1024} for $N_M$, $I$, and $D_p$, respectively, while strictly controlling for look-back window and prediction length. Specifically, we examine the impact of four different prediction lengths, *i.e.*, 512 → {96, 192, 336, 720}, and present the results in Figure 9. Figure 9a illustrates that, as the number of masked series increases, the forecasting performance initially improves due to the capture of more decomposed frequency patterns in the two energy branches, but eventually decreases due to information redundancy. Figure 9b shows that increasing the prompt number results in minimal changes in forecasting performance, with the optimal result achieved when the prompt number is set to 128. Insufficient prompts fail to capture enough domain information, while too many introduce redundancy, hindering the model's efficient adaptation to new and unseen datasets. Figure 9c demonstrates that the forecasting performance fluctuates slightly before reaching 256, but it drops sharply thereafter. For small datasets such as ETTm1/m2, a compact latent dimension suffices to capture the data's underlying characteristics; therefore, we opt for a dimension of 256 to strike an optimal balance between model capacity and expressive power.

### D.3 Scaling Analysis

To investigate the scalability of SEMPO, we scaled both model size and dataset size and evaluated its predictive performance on two ETTh datasets. As shown in Figure 10 **Left**, expanding the dataset size consistently improves model performance, in line with the scaling law. As the dataset size increases, the performance improvement plateaus, indicating diminishing returns from further expansion. However, the performance remains consistently below the MSE of the SOTA FMs $Moirai_S$ and $Moirai_B$ with a dataset size of 27B. Regarding model size, as seen in Figure 10 **Right**, SEMPO follows a similar trend. Performance declines in the later stages,

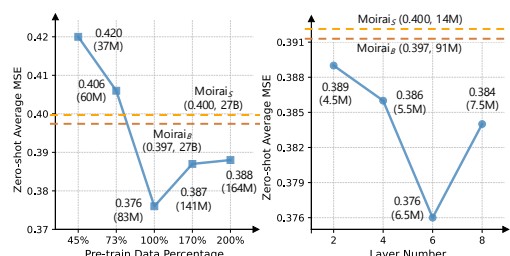

Figure 10: Scalability analysis on ETTh1/h2.

likely due to a mismatch between model capacity and dataset size. Despite this, SEMPO continues to surpass $Moirai_S$ and $Moirai_B$ with 14M and 91M parameters, respectively.

### D.4 Efficiency Analysis

Since many Transformer-based FMs process data in a purely univariate fashion, the dataset size and inference costs are directly proportional. To further highlight the performance and efficiency benefits, in addition to the smaller ETTh1 dataset (see Figure 6), we also compare the inference costs of SEMPO and other FMs on the Weather and ETTm2 datasets. All experiments are conducted using a batch size of 32 on a single A6000 48G GPU, as higher batch sizes often lead to out-of-memory (OOM) errors in many FMs, such as $Chronos_L$. The zero-shot results are shown in Figure 11, where SEMPO significantly outperforms all other FMs in both efficiency and accuracy.

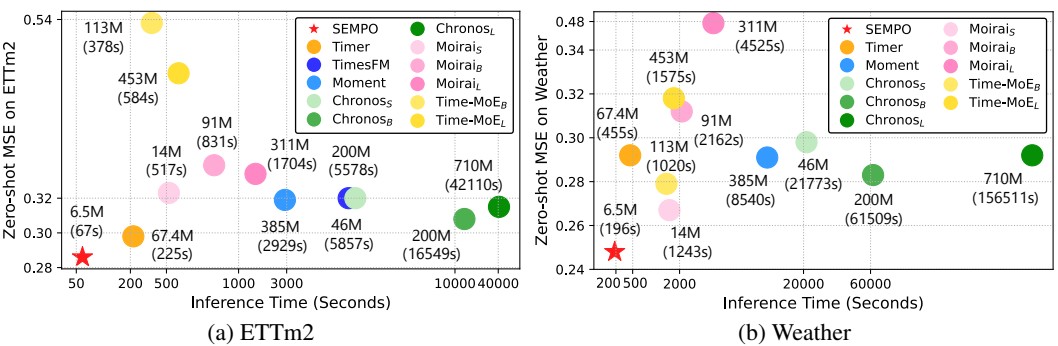

(a) ETTm2            (b) Weather

Figure 11: Additional efficiency comparison on the ETTm2 and Weather datasets.

Beyond the comparisons among Transformer-based FMs, we have additionally included an efficiency comparison with TTM on the ETTh1 and ETTm2 datasets, summarized in Table 7 right. While SEMPO is less competitive in parameter size and inference time due to its more complex Transformer-based architecture, it uses only 83M pre-training data (vs. TTM's 1B) and still achieves lower MSE on both datasets. Moreover, SEMPO outperforms TTM in 31 out of 50 cases across five diverse datasets (see Table 10 in Appendix D.6), with par-

Table 7: Efficiency comparison with TTM on the ETTh1 and ETTm2 datasets. **Red**: the best.

| Datasets | Models | Inference Time (Seconds) | Parameters (M) | MSE |
|---|---|---|---|---|
| ETTh1 | TTM | **3.2** | **1** | 0.411 |
|  | SEMPO | 22 | 6.5 | **0.410** |
| ETTm2 | TTM | **5.2** | **1** | 0.288 |
|  | SEMPO | 67 | 6.5 | **0.286** |

ticularly substantial advantages on large-scale datasets such as Traffic and Electricity, demonstrating stronger knowledge transfer capability and superior sample efficiency.

## D.5 Visualization of Predictions

To provide a clearer comparison between SEMPO and other foundation models, we present prediction case studies on the ETTh2 dataset, as illustrated in Figure 12. We select several representative time series FMs as baselines, including $\text{Moirai}_L$, Timer, $\text{Time-MoE}_L$, $\text{Time-MoE}_B$, and $\text{Chronos}_L$. SEMPO consistently delivers more accurate forecasting of future series variations compared to these diverse FMs, underscoring its superior predictive capability.

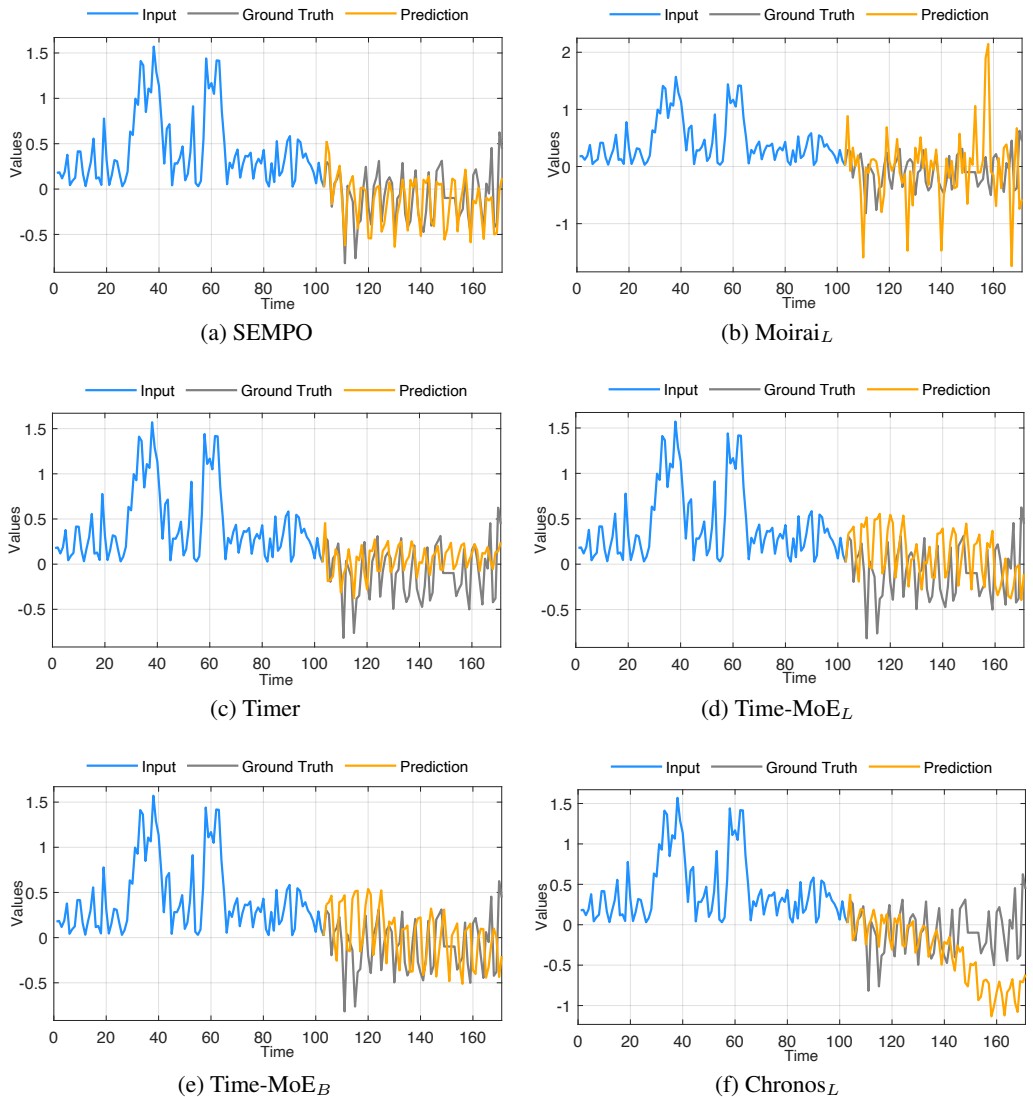

Figure 12: Visualization of zero-shot prediction cases on the ETTh2 dataset by various FMs, using lookback window length $L = 512$ and prediction length $H = 336$.

## D.6 Full Results Tables

Here, we present the comprehensive versions of the tables in the main text. These versions include the full results for multiple prediction lengths across all datasets. To save space in the main paper, some of these results are averaged across different prediction lengths.

**Full Results for Zero-shot Forecasting.** Table 8 and Table 9 report the zero-shot results on the TSLib and GIFT-Eval benchmarks, respectively. It is observed that with only 6.5M parameters and using 83M pre-training data, SEMPO consistently outperforms other time series FMs on the

Table 8: Full zero-shot results on the TSLib benchmark [25]. Lower MSE/MAE values indicate better prediction. '-' denotes dataset used in pre-training and excluded in the evaluation. Values in ( , ) denote model size and pre-training data size, respectively. 'S': Small, 'B': Base, 'L': Large. **Red**: the best, **Blue**: the second best.

| Models | Metrics | SEMPO (6.5M,83M) | | Time-MoE$_B$ (113M,309B) | | Time-MoE$_L$ (453M,309B) | | Timer (67.4M,28B) | | Moirai$_S$ (14M,27B) | | Moirai$_B$ (91M,27B) | | Moirai$_L$ (311M,27B) | | Chronos$_S$ (46M,84B) | | Chronos$_B$ (200M,84B) | | Chronos$_L$ (710M,84B) | | TimesFM (200M,100B) | | Moment (385M,1.13B) | |
|---|---|---|---|---|---|---|---|---|---|---|---|---|---|---|---|---|---|---|---|---|---|---|---|---|---|
| | | MSE | MAE | MSE | MAE | MSE | MAE | MSE | MAE | MSE | MAE | MSE | MAE | MSE | MAE | MSE | MAE | MSE | MAE | MSE | MAE | MSE | MAE | MSE | MAE |
| ETTh1 | 96 | **0.384** | 0.408 | 0.358 | **0.382** | **0.350** | **0.382** | 0.414 | 0.439 | 0.407 | 0.405 | 0.394 | **0.399** | 0.410 | 0.403 | 0.476 | 0.411 | 0.452 | 0.396 | 0.453 | 0.401 | 0.432 | 0.405 | 0.706 | 0.561 |
| | 192 | **0.409** | 0.426 | **0.404** | **0.417** | **0.402** | **0.423** | 0.440 | 0.455 | 0.443 | 0.425 | 0.431 | 0.430 | 0.459 | 0.434 | 0.547 | 0.452 | 0.513 | 0.429 | 0.525 | 0.428 | 0.492 | 0.438 | 0.716 | 0.579 |
| | 336 | **0.417** | **0.433** | 0.449 | 0.454 | **0.447** | 0.459 | 0.455 | 0.463 | 0.465 | 0.438 | 0.450 | **0.437** | 0.486 | 0.453 | 0.589 | 0.481 | 0.553 | 0.450 | 0.570 | 0.452 | 0.519 | 0.458 | 0.705 | 0.583 |
| | 720 | **0.432** | **0.454** | 0.570 | 0.543 | 0.543 | 0.534 | 0.496 | 0.496 | 0.475 | 0.461 | **0.456** | **0.457** | 0.511 | 0.483 | 0.592 | 0.509 | 0.578 | 0.484 | 0.619 | 0.493 | 0.512 | 0.477 | 0.705 | 0.597 |
| | Avg | **0.410** | **0.430** | 0.445 | 0.449 | 0.435 | 0.449 | 0.451 | 0.463 | 0.448 | 0.432 | **0.433** | **0.431** | 0.466 | 0.443 | 0.551 | 0.463 | 0.524 | 0.439 | 0.541 | 0.443 | 0.489 | 0.444 | 0.708 | 0.580 |
| ETTh2 | 96 | **0.282** | **0.342** | 0.302 | 0.356 | 0.301 | 0.354 | 0.305 | 0.355 | 0.288 | 0.345 | **0.285** | **0.342** | 0.294 | **0.344** | 0.309 | 0.354 | 0.307 | 0.352 | 0.297 | 0.344 | 0.311 | 0.345 | 0.373 | 0.416 |
| | 192 | **0.334** | **0.384** | 0.438 | 0.431 | 0.407 | 0.417 | 0.365 | 0.406 | **0.352** | 0.396 | **0.352** | 0.391 | 0.368 | **0.387** | 0.386 | 0.399 | 0.388 | 0.388 | 0.379 | 0.392 | 0.401 | 0.397 | 0.384 | 0.422 |
| | 336 | **0.355** | **0.403** | 0.617 | 0.509 | 0.514 | 0.476 | 0.378 | 0.413 | **0.375** | 0.420 | 0.384 | 0.418 | 0.403 | **0.411** | 0.428 | 0.426 | 0.432 | 0.422 | 0.414 | 0.412 | 0.436 | 0.430 | 0.386 | 0.426 |
| | 720 | **0.395** | **0.435** | 0.907 | 0.622 | 0.689 | 0.561 | 0.414 | 0.457 | **0.405** | **0.442** | 0.418 | 0.446 | 0.465 | 0.453 | 0.454 | 0.455 | 0.442 | 0.443 | 0.451 | 0.452 | 0.437 | 0.450 | 0.425 | 0.454 |
| | Avg | **0.341** | **0.391** | 0.566 | 0.479 | 0.477 | 0.452 | 0.366 | 0.408 | **0.355** | 0.401 | 0.360 | 0.399 | 0.382 | **0.397** | 0.394 | 0.409 | 0.392 | 0.401 | 0.385 | 0.400 | 0.396 | 0.405 | 0.392 | 0.430 |
| ETTm1 | 96 | 0.466 | 0.443 | **0.339** | **0.368** | **0.309** | **0.357** | 0.440 | 0.422 | 0.509 | 0.458 | 0.528 | 0.433 | 0.532 | 0.433 | 0.508 | 0.421 | 0.447 | 0.400 | 0.451 | 0.396 | 0.366 | 0.374 | 0.679 | 0.544 |
| | 192 | 0.484 | 0.455 | 0.443 | 0.440 | **0.425** | **0.435** | 0.505 | 0.458 | 0.541 | 0.471 | 0.549 | 0.448 | 0.577 | 0.455 | 0.606 | 0.474 | 0.535 | 0.451 | 0.464 | 0.428 | **0.413** | **0.401** | 0.690 | 0.550 |
| | 336 | **0.506** | **0.469** | 0.532 | 0.502 | 0.516 | 0.495 | 0.570 | 0.490 | 0.563 | 0.479 | 0.579 | 0.473 | 0.620 | 0.478 | 0.663 | 0.506 | 0.602 | 0.485 | 0.541 | 0.487 | **0.445** | **0.429** | 0.701 | 0.557 |
| | 720 | **0.557** | **0.498** | 0.716 | 0.610 | 0.683 | 0.597 | 0.659 | 0.534 | 0.604 | 0.499 | 0.606 | 0.503 | 0.674 | 0.508 | 0.734 | 0.545 | 0.682 | 0.526 | 0.630 | 0.507 | **0.513** | **0.470** | 0.719 | 0.569 |
| | Avg | 0.503 | 0.466 | 0.507 | 0.480 | **0.483** | 0.471 | 0.544 | 0.476 | 0.554 | 0.477 | 0.566 | 0.464 | 0.601 | 0.468 | 0.628 | 0.487 | 0.566 | 0.465 | 0.521 | **0.448** | **0.434** | **0.419** | 0.697 | 0.555 |
| ETTm2 | 96 | **0.196** | 0.286 | 0.197 | 0.287 | 0.197 | 0.285 | 0.203 | 0.285 | 0.228 | 0.296 | 0.227 | 0.290 | 0.223 | 0.288 | 0.209 | 0.287 | 0.204 | **0.280** | 0.206 | 0.283 | **0.189** | **0.257** | 0.230 | 0.308 |
| | 192 | **0.252** | **0.323** | 0.338 | 0.381 | 0.329 | 0.376 | **0.265** | 0.327 | 0.275 | 0.324 | 0.306 | 0.334 | 0.303 | 0.331 | 0.280 | **0.321** | 0.279 | 0.330 | 0.277 | 0.325 | 0.285 | 0.338 | 0.285 | 0.338 |
| | 336 | **0.306** | **0.354** | 0.586 | 0.501 | 0.534 | 0.481 | **0.319** | 0.361 | 0.335 | **0.360** | 0.366 | 0.373 | 0.354 | 0.364 | 0.344 | 0.371 | 0.327 | 0.362 | 0.339 | 0.365 | 0.350 | 0.381 | 0.339 | 0.369 |
| | 720 | **0.391** | **0.404** | 1.034 | 0.683 | 0.978 | 0.668 | **0.405** | **0.410** | 0.453 | 0.425 | 0.456 | 0.429 | 0.456 | 0.423 | 0.448 | 0.430 | 0.432 | 0.415 | 0.437 | 0.420 | 0.464 | 0.448 | 0.423 | 0.424 |
| | Avg | **0.286** | **0.341** | 0.538 | 0.463 | 0.509 | 0.452 | **0.298** | 0.346 | 0.323 | 0.351 | 0.339 | 0.356 | 0.334 | 0.352 | 0.320 | 0.355 | 0.308 | **0.344** | 0.315 | 0.350 | 0.320 | 0.353 | 0.319 | 0.360 |
| Weather | 96 | **0.171** | **0.228** | **0.159** | **0.213** | **0.159** | **0.214** | 0.190 | 0.236 | 0.180 | 0.229 | 0.208 | 0.221 | 0.213 | **0.213** | 0.212 | 0.238 | 0.198 | 0.231 | 0.207 | 0.232 | - | - | 0.216 | 0.271 |
| | 192 | **0.218** | 0.269 | **0.214** | **0.267** | **0.217** | 0.271 | 0.261 | 0.293 | 0.231 | 0.279 | 0.281 | 0.270 | 0.342 | **0.262** | 0.266 | 0.283 | 0.248 | 0.276 | 0.260 | 0.277 | - | - | 0.264 | 0.306 |
| | 336 | **0.267** | **0.304** | 0.292 | 0.326 | 0.312 | 0.343 | 0.332 | 0.340 | **0.289** | 0.330 | 0.340 | 0.313 | 0.527 | 0.352 | 0.321 | 0.321 | 0.301 | 0.314 | 0.310 | 0.313 | - | - | 0.313 | 0.336 |
| | 720 | **0.336** | **0.350** | 0.453 | 0.430 | 0.586 | 0.508 | 0.385 | 0.381 | **0.367** | 0.385 | 0.420 | 0.376 | 0.826 | 0.369 | 0.393 | **0.367** | 0.386 | 0.361 | 0.390 | 0.365 | - | - | 0.369 | 0.380 |
| | Avg | **0.248** | **0.287** | 0.279 | 0.309 | 0.318 | 0.334 | 0.292 | 0.312 | **0.267** | 0.306 | 0.312 | 0.295 | 0.477 | **0.289** | 0.298 | 0.302 | 0.283 | 0.295 | 0.292 | 0.297 | - | - | 0.291 | 0.323 |
| Electricity | 96 | **0.168** | 0.271 | - | - | - | - | 0.210 | 0.312 | 0.212 | 0.304 | **0.169** | **0.269** | 0.193 | 0.275 | 0.199 | **0.267** | 0.196 | 0.273 | 0.198 | 0.278 | - | - | 0.844 | 0.761 |
| | 192 | **0.183** | **0.283** | - | - | - | - | 0.239 | 0.337 | 0.224 | 0.315 | **0.186** | **0.285** | **0.186** | 0.296 | 0.218 | 0.289 | 0.207 | **0.285** | 0.214 | 0.294 | - | - | 0.850 | 0.762 |
| | 336 | **0.198** | **0.297** | - | - | - | - | 0.284 | 0.372 | 0.244 | 0.331 | **0.215** | **0.299** | 0.221 | 0.311 | 0.244 | 0.321 | 0.238 | 0.314 | 0.239 | 0.314 | - | - | 0.862 | 0.766 |
| | 720 | **0.238** | **0.329** | - | - | - | - | 0.456 | 0.479 | 0.291 | 0.365 | **0.257** | **0.332** | 0.296 | 0.355 | 0.324 | 0.371 | 0.310 | 0.355 | 0.316 | 0.362 | - | - | 0.888 | 0.774 |
| | Avg | **0.196** | **0.295** | - | - | - | - | 0.297 | 0.375 | 0.243 | 0.329 | **0.207** | **0.296** | 0.224 | 0.309 | 0.246 | 0.312 | 0.336 | 0.329 | 0.326 | 0.328 | - | - | 0.861 | 0.766 |
| Traffic | 96 | **0.441** | **0.333** | - | - | - | - | **0.526** | **0.368** | - | - | - | - | - | - | 0.562 | 0.378 | 0.558 | 0.375 | 0.541 | 0.364 | - | - | 1.390 | 0.800 |
| | 192 | **0.456** | **0.339** | - | - | - | - | **0.561** | **0.385** | - | - | - | - | - | - | 0.579 | 0.412 | 0.560 | 0.399 | 0.570 | 0.406 | - | - | 1.403 | 0.802 |
| | 336 | **0.467** | **0.344** | - | - | - | - | 0.614 | **0.412** | - | - | - | - | - | - | **0.594** | 0.420 | 0.584 | 0.413 | 0.571 | 0.404 | - | - | 1.415 | 0.804 |
| | 720 | **0.503** | **0.360** | - | - | - | - | 0.749 | 0.464 | - | - | - | - | - | - | **0.723** | 0.472 | 0.711 | 0.464 | 0.719 | 0.469 | - | - | 1.437 | 0.808 |
| | Avg | **0.466** | **0.344** | - | - | - | - | **0.613** | **0.407** | - | - | - | - | - | - | 0.614 | 0.420 | 0.603 | 0.413 | 0.600 | 0.411 | - | - | 1.411 | 0.804 |
| 1$^{st}$ Count | | 49 | | | 12 | | | | 0 | | | 3 | | | | 1 | | | | | | 10 | | | 0 |

majority of datasets. Given that the GIFT-Eval benchmark includes short- (*i.e.*, $H = L$), medium- (*i.e.*, $H = 10 \times L$), and long-term (*i.e.*, $H = 15 \times L$) prediction lengths, the results in Table 9 further demonstrate SEMPO's strong adaptability to a wide range of prediction horizons, achieving outstanding forecasting performance on most datasets.

**Full Results for Zero-shot Comparisons with TTM.** Since Table 8 provides the comparison with Transformer-based time series FMs, Table 10 further reports results the comparison to the lightweight MLP-based FM, TTM. Existing research [27] has experimentally demonstrated that simpler structures, such as MLPs, may be more suitable for smaller datasets, while larger datasets, due to their complex relationships, require more contextual structures, such as Transformers. To ensure a fair comparison, we exclude two small datasets, ETTh1 and ETTm1, from the TSLib benchmark to maintain a balanced dataset ratio for evaluation. The results are shown in Table 10. Despite TTM having only 1M parameters, SEMPO outperforms in the majority of cases (31 out of 50) utilizing only 83M data, showcasing its better capabilities in knowledge transfer than TTM. Notably, SEMPO demonstrates substantial advantages on large datasets, such as Traffic and Electricity, which is consistent with our earlier analysis.

**Full Results for Zero-shot Comparisons with Chronos-Bolt and TimesFM (500M).** Since new versions of Chronos and TimesFM are available, we have added zero-shot results for the latest Chronos-Bolt and TimesFM (500M), summarized in Table 11 for an up-to-date comparison. Across

Table 9: Full zero-shot results on the GIFT-Eval benchmark [49]. Lower NRMSE/SMAPE values indicate better prediction. '$S$': Small, '$B$': Base, '$L$': Large. **Red**: the best. Abbr. stands for abbreviation.

| Datasets | Metrics | SEMPO | Chronos$_L$ | Chronos$_B$ | Chronos$_S$ |
|---|---|---|---|---|---|
| m_dense | NRMSE | 0.392 | 0.319 | **0.314** | 0.317 |
| | SMAPE | 0.295 | 0.186 | **0.181** | 0.183 |
| loop_seattle (abbr. loop) | NRMSE | **0.160** | 0.175 | 0.177 | 0.181 |
| | SMAPE | **0.131** | 0.132 | 0.133 | 0.134 |
| sz_taxi (abbr. taxi) | NRMSE | **0.369** | 0.372 | 0.375 | 0.380 |
| | SMAPE | **0.406** | 0.425 | 0.418 | 0.450 |
| solar | NRMSE | **1.084** | 1.144 | 1.086 | 1.144 |
| | SMAPE | **1.217** | 1.270 | 1.230 | 1.252 |
| bizitobs_application (abbr. biz_app) | NRMSE | 0.213 | 0.160 | 0.155 | **0.147** |
| | SMAPE | 0.162 | **0.098** | 0.110 | 0.107 |
| bizitobs_l2c (abbr. bb_12c) | NRMSE | **0.780** | 1.010 | 1.040 | 0.999 |
| | SMAPE | **0.880** | 1.140 | 1.135 | 1.076 |
| bitbrains_service (abbr. bb_svc) | NRMSE | 0.299 | 0.236 | 0.233 | **0.227** |
| | SMAPE | 0.228 | **0.139** | 0.150 | 0.140 |
| car_parts (abbr. car) | MASE | 3.075 | 3.013 | 3.008 | **2.958** |
| | SMAPE | **1.796** | 1.862 | 1.871 | 1.871 |
| jena_weather (abbr. jena) | NRMSE | **0.236** | 0.241 | 0.243 | 0.261 |
| | SMAPE | **0.638** | 0.657 | 0.653 | 0.649 |

Table 10: Zero-shot comparison with TTM on the ETTh2/m2, Weather, Electricity, and Traffic datasets. Lower MSE/MAE values indicate better prediction. Values in ( , ) denote model size and pre-training data size, respectively. **Red**: the best.

| Datasets | | ETTh2 | | ETTm2 | | Weather | | Electricity | | Traffic | | $1^{st}$ Count |
|---|---|---|---|---|---|---|---|---|---|---|---|---|
| Metrics | | MSE | MAE | MSE | MAE | MSE | MAE | MSE | MAE | MSE | MAE | |
| TTM (1M, 1B) | 96 | **0.276** | **0.335** | **0.177** | **0.259** | **0.151** | **0.196** | 0.182 | 0.274 | 0.508 | 0.342 | |
| | 192 | 0.346 | **0.383** | **0.246** | **0.307** | **0.195** | **0.241** | 0.196 | 0.287 | 0.538 | 0.355 | |
| | 336 | 0.385 | 0.417 | 0.324 | **0.350** | **0.256** | **0.285** | 0.219 | 0.307 | 0.574 | 0.373 | 19 |
| | 720 | 0.420 | 0.450 | 0.406 | 0.405 | **0.319** | **0.333** | 0.261 | 0.342 | 0.617 | 0.390 | |
| | Avg | 0.357 | 0.396 | 0.288 | **0.330** | **0.230** | **0.264** | 0.215 | 0.303 | 0.559 | 0.365 | |
| SEMPO (6.5M, 83M) | 96 | 0.282 | 0.342 | 0.196 | 0.286 | 0.171 | 0.228 | **0.168** | **0.271** | **0.441** | **0.333** | |
| | 192 | **0.334** | 0.384 | 0.252 | 0.323 | 0.218 | 0.269 | **0.183** | **0.283** | **0.456** | **0.339** | |
| | 336 | **0.355** | **0.403** | **0.306** | 0.354 | 0.267 | 0.304 | **0.198** | **0.297** | **0.467** | **0.344** | 31 |
| | 720 | **0.395** | **0.435** | **0.391** | **0.404** | 0.336 | 0.350 | **0.238** | **0.329** | **0.503** | **0.360** | |
| | Avg | **0.341** | **0.391** | **0.286** | 0.341 | 0.248 | 0.287 | **0.196** | **0.295** | **0.466** | **0.344** | |

five datasets, Chronos-Bolt achieves the best performance in 20 cases, SEMPO in 16, and TimesFM (500M) in 7. We can observe that SEMPO significantly outperforms TimesFM (500M). It should be noted that, even with the small version of Chronos-Bolt (*i.e.*, Chronos-Bolt$_S$ ), it is trained on nearly 100B time points with 48M parameters. In contrast, SEMPO is trained on only 83M points with just 6.5M parameters, yet it still ranks in second in terms of the overall performance. This compelling data efficiency–performance balance highlights SEMPO's promise as an effective lightweight FM for resource-constrained environments, to which the other FMs such as Chronos-Bolt and TimesFM (500M) are infeasible.

**Full Results for Different Lookback Window Lengths.** Given that existing time series FMs, such as Time-MoE and TTM, utilize larger lookback window lengths, we additionally pre-train two variants: SEMPO-Enhanced (SEMPO$_E$) with $L = 1024, L_p = 128$ (parameters: 7.3M), and SEMPO-Advanced (SEMPO$_A$) with $L = 1536, L_p = 128$ (parameters: 9.9M). The default configuration, with $L = 512, L_p = 64$ (parameters: 6.5M), is denoted as SEMPO$_B$. To avoid data leakage, we exclude

Table 11: Zero-shot comparison with Chronos-Bolt and TimesFM (500M) on the ETTh1/h2/m1/m2 and Weather datasets. Lower MSE/MAE values indicate better prediction. Values in ( , ) denote model size and pre-training data size, respectively. **Red**: the best.

| Datasets | | ETTh1 | | ETTh2 | | ETTm1 | | ETTm2 | | Weather | | $1^{st}$ Count |
|---|---|---|---|---|---|---|---|---|---|---|---|---|
| Metrics | | MSE | MAE | MSE | MAE | MSE | MAE | MSE | MAE | MSE | MAE | |
| TimesFM (500M, >100B) | 192 | 0.428 | **0.409** | 0.370 | 0.385 | 0.388 | 0.376 | 0.268 | 0.307 | - | - | |
| | 336 | 0.451 | **0.424** | 0.416 | 0.420 | **0.426** | 0.402 | 0.327 | 0.347 | - | - | 7 |
| | 720 | 0.463 | **0.447** | 0.443 | 0.448 | **0.491** | 0.444 | 0.421 | 0.407 | - | - | |
| | Avg | 0.447 | **0.427** | 0.410 | 0.418 | **0.435** | 0.407 | 0.339 | 0.354 | - | - | |
| Chronos-Bolt$_S$ (48M, 100B) | 192 | 0.451 | 0.415 | 0.351 | **0.371** | 0.382 | **0.366** | **0.239** | **0.291** | 0.218 | **0.250** | |
| | 336 | 0.497 | 0.437 | 0.396 | 0.405 | 0.433 | 0.396 | 0.306 | **0.334** | 0.275 | **0.290** | 20 |
| | 720 | 0.499 | 0.456 | 0.413 | 0.429 | 0.530 | 0.447 | 0.413 | 0.398 | 0.354 | **0.338** | |
| | Avg | 0.482 | 0.436 | 0.387 | 0.402 | 0.448 | 0.403 | 0.319 | **0.341** | 0.282 | **0.293** | |
| Chronos-Bolt$_B$ (205M, 100B) | 192 | 0.435 | 0.412 | 0.355 | **0.371** | **0.381** | **0.366** | 0.247 | 0.295 | 0.222 | 0.258 | |
| | 336 | 0.476 | 0.433 | 0.400 | **0.403** | 0.432 | **0.393** | 0.314 | 0.336 | 0.282 | 0.298 | |
| | 720 | 0.480 | 0.449 | 0.414 | **0.422** | 0.525 | **0.439** | 0.415 | **0.395** | 0.366 | 0.348 | |
| | Avg | 0.464 | 0.431 | 0.390 | **0.399** | 0.446 | **0.399** | 0.325 | 0.342 | 0.290 | 0.301 | |
| SEMPO (6.5M, 83M) | 192 | **0.409** | 0.426 | **0.334** | 0.384 | 0.484 | 0.455 | 0.252 | 0.323 | **0.216** | 0.269 | |
| | 336 | **0.417** | 0.433 | **0.355** | **0.403** | 0.506 | 0.469 | **0.305** | 0.354 | **0.267** | 0.304 | 16 |
| | 720 | **0.432** | 0.454 | **0.395** | 0.435 | 0.557 | 0.498 | **0.391** | 0.404 | **0.336** | 0.350 | |
| | Avg | **0.419** | 0.438 | **0.361** | 0.407 | 0.516 | 0.474 | **0.316** | 0.360 | **0.273** | 0.308 | |

Table 12: Zero-shot comparison with Time-MoE on the TSLib benchmark, excluding Electricity and Traffic datasets, under different lookback widnow lengths. Lower MSE/MAE values indicate better prediction. '$B$': Base, '$E$': Enhanced, '$A$': Advanced, '$L$': Large, '$*$': $L = 1024$, '$\dagger$': $L = 1536$. Values in ( , ) denote model size and pre-training data size, respectively. **Red**: the best.

| Models | | SEMPO$_B$ (6.5M, 83M) | | SEMPO$_E$ (7.3M, 83M) | | SEMPO$_A$ (9.9M, 83M) | | Time-MoE$_L^*$ (453M, 309B) | | Time-MoE$_B^*$ (113M, 309B) | | Time-MoE$_L^\dagger$ (453M, 309B) | | Time-MoE$_B^\dagger$ (113M, 309B) | |
|---|---|---|---|---|---|---|---|---|---|---|---|---|---|---|---|
| Metrics | | MSE | MAE | MSE | MAE | MSE | MAE | MSE | MAE | MSE | MAE | MSE | MAE | MSE | MAE |
| ETTh1 | 96 | 0.384 | 0.408 | 0.392 | 0.419 | 0.392 | 0.423 | 0.339 | 0.375 | 0.343 | **0.373** | **0.338** | 0.375 | 0.343 | 0.375 |
| | 192 | 0.409 | 0.426 | 0.415 | 0.433 | 0.422 | 0.443 | 0.389 | 0.412 | **0.385** | **0.405** | 0.391 | 0.415 | 0.387 | 0.407 |
| | 336 | **0.417** | **0.433** | 0.423 | 0.440 | 0.435 | 0.452 | 0.429 | 0.443 | 0.421 | 0.437 | 0.428 | 0.445 | 0.420 | 0.438 |
| | 720 | **0.432** | **0.454** | 0.440 | 0.465 | 0.447 | 0.468 | 0.498 | 0.504 | 0.507 | 0.511 | 0.482 | 0.497 | 0.493 | 0.507 |
| | Avg | 0.410 | **0.430** | 0.417 | 0.439 | 0.424 | 0.446 | 0.414 | 0.433 | 0.414 | 0.431 | **0.409** | 0.433 | 0.410 | 0.431 |
| ETTh2 | 96 | 0.282 | 0.342 | 0.309 | 0.365 | 0.308 | 0.362 | 0.286 | 0.338 | 0.271 | 0.332 | 0.277 | 0.335 | **0.266** | **0.329** |
| | 192 | 0.334 | 0.384 | 0.362 | 0.400 | 0.370 | 0.405 | 0.364 | 0.385 | 0.348 | 0.383 | 0.359 | 0.385 | **0.341** | **0.381** |
| | 336 | **0.355** | **0.403** | 0.378 | 0.415 | 0.388 | 0.420 | 0.413 | 0.420 | 0.393 | 0.416 | 0.411 | 0.423 | 0.385 | 0.414 |
| | 720 | **0.395** | 0.435 | 0.399 | **0.417** | 0.409 | 0.440 | 0.463 | 0.460 | 0.433 | 0.453 | 0.492 | 0.478 | 0.412 | 0.448 |
| | Avg | **0.341** | **0.391** | 0.362 | 0.399 | 0.368 | 0.406 | 0.381 | 0.400 | 0.361 | 0.396 | 0.384 | 0.405 | 0.351 | 0.393 |
| ETTm1 | 96 | 0.466 | 0.443 | 0.454 | 0.435 | 0.455 | 0.447 | 0.253 | 0.319 | 0.266 | 0.327 | **0.232** | **0.308** | 0.240 | 0.315 |
| | 192 | 0.484 | 0.455 | 0.473 | 0.453 | 0.471 | 0.462 | 0.347 | 0.381 | 0.353 | 0.388 | **0.314** | **0.364** | 0.318 | 0.370 |
| | 336 | 0.506 | 0.469 | 0.491 | 0.464 | 0.482 | 0.470 | 0.428 | 0.436 | 0.441 | 0.448 | **0.392** | **0.416** | 0.402 | 0.425 |
| | 720 | 0.557 | 0.498 | 0.527 | 0.487 | **0.495** | **0.479** | 0.564 | 0.525 | 0.616 | 0.555 | 0.517 | 0.499 | 0.559 | 0.524 |
| | Avg | 0.503 | 0.466 | 0.486 | 0.459 | 0.475 | 0.464 | 0.398 | 0.415 | 0.419 | 0.429 | **0.363** | **0.396** | 0.379 | 0.408 |
| ETTm2 | 96 | 0.196 | 0.286 | 0.194 | 0.284 | 0.191 | 0.283 | 0.170 | 0.260 | 0.170 | 0.264 | 0.167 | 0.258 | **0.165** | **0.257** |
| | 192 | 0.252 | 0.323 | 0.245 | 0.318 | 0.247 | 0.320 | 0.250 | 0.321 | 0.254 | 0.330 | 0.246 | 0.317 | **0.237** | **0.315** |
| | 336 | 0.306 | 0.354 | 0.292 | 0.347 | **0.286** | **0.345** | 0.341 | 0.381 | 0.378 | 0.407 | 0.338 | 0.377 | 0.336 | 0.380 |
| | 720 | 0.391 | 0.404 | 0.372 | 0.395 | **0.360** | **0.390** | 0.522 | 0.483 | 0.597 | 0.524 | 0.531 | 0.483 | 0.517 | 0.483 |
| | Avg | 0.286 | 0.341 | 0.275 | 0.336 | **0.271** | **0.334** | 0.320 | 0.361 | 0.349 | 0.381 | 0.320 | 0.358 | 0.313 | 0.358 |
| Weather | 96 | 0.171 | 0.228 | 0.174 | 0.231 | 0.177 | 0.234 | 0.153 | 0.206 | 0.153 | 0.204 | **0.150** | 0.202 | 0.151 | **0.201** |
| | 192 | 0.218 | 0.269 | 0.217 | 0.268 | 0.231 | 0.278 | 0.216 | 0.267 | 0.210 | 0.260 | 0.206 | 0.258 | **0.203** | **0.253** |
| | 336 | 0.267 | 0.304 | **0.261** | **0.299** | 0.273 | 0.307 | 0.315 | 0.341 | 0.287 | 0.318 | 0.288 | 0.323 | 0.270 | 0.306 |
| | 720 | 0.336 | 0.350 | **0.325** | **0.343** | 0.334 | 0.346 | 0.566 | 0.489 | 0.433 | 0.411 | 0.489 | 0.453 | 0.395 | 0.391 |
| | Avg | 0.248 | 0.287 | **0.244** | **0.285** | 0.253 | 0.291 | 0.312 | 0.325 | 0.270 | 0.298 | 0.283 | 0.309 | 0.254 | 0.287 |
| $1^{st}$ Count | | 25 | | | | | | 25 | | | | | | | |

Table 13: Full-shot results on the ETTh2, ETTm2, Weather and Traffic datasets. Lower MSE/MAE values indicate better prediction. **Red**: the best.

| Models | | SEMPO | | GPT4TS | | S$^2$P-LLM | | iTransformer | | DLinear | | PatchTST | | TimesNet | |
|---|---|---|---|---|---|---|---|---|---|---|---|---|---|---|---|
| Metrics | | MSE | MAE | MSE | MAE | MSE | MAE | MSE | MAE | MSE | MAE | MSE | MAE | MSE | MAE |
| ETTh2 | 96 | **0.273** | **0.334** | 0.285 | 0.342 | 0.278 | 0.340 | 0.297 | 0.348 | 0.302 | 0.368 | 0.274 | 0.337 | 0.351 | 0.399 |
| | 192 | 0.333 | **0.376** | 0.354 | 0.389 | **0.246** | 0.385 | 0.371 | 0.403 | 0.404 | 0.433 | 0.341 | 0.382 | 0.394 | 0.429 |
| | 336 | 0.355 | 0.400 | 0.373 | 0.407 | 0.367 | 0.406 | 0.404 | 0.428 | 0.511 | 0.498 | **0.329** | **0.384** | 0.415 | 0.443 |
| | 720 | 0.399 | 0.435 | 0.406 | 0.441 | 0.400 | 0.436 | 0.424 | 0.444 | 0.815 | 0.640 | **0.379** | **0.422** | 0.477 | 0.481 |
| | Avg | 0.340 | 0.386 | 0.355 | 0.395 | **0.323** | 0.392 | 0.374 | 0.406 | 0.508 | 0.485 | 0.331 | **0.381** | 0.409 | 0.438 |
| ETTm2 | 96 | **0.160** | **0.251** | 0.173 | 0.262 | 0.165 | 0.257 | 0.175 | 0.266 | 0.164 | 0.255 | 0.166 | 0.256 | 0.233 | 0.305 |
| | 192 | **0.221** | **0.294** | 0.229 | 0.301 | 0.222 | 0.299 | 0.242 | 0.312 | 0.224 | 0.304 | 0.223 | 0.296 | 0.265 | 0.328 |
| | 336 | **0.273** | **0.328** | 0.286 | 0.341 | 0.277 | 0.330 | 0.282 | 0.340 | 0.277 | 0.339 | 0.274 | 0.329 | 0.379 | 0.392 |
| | 720 | **0.349** | **0.380** | 0.378 | 0.401 | 0.363 | 0.390 | 0.378 | 0.398 | 0.371 | 0.401 | 0.362 | 0.385 | 0.390 | 0.407 |
| | Avg | **0.251** | **0.313** | 0.266 | 0.326 | 0.257 | 0.319 | 0.269 | 0.329 | 0.259 | 0.325 | 0.256 | 0.317 | 0.317 | 0.358 |
| Weather | 96 | **0.143** | **0.193** | 0.162 | 0.212 | 0.145 | 0.195 | 0.159 | 0.208 | 0.170 | 0.230 | 0.149 | 0.198 | 0.193 | 0.244 |
| | 192 | **0.189** | **0.234** | 0.204 | 0.248 | 0.190 | 0.235 | 0.200 | 0.248 | 0.212 | 0.267 | 0.194 | 0.241 | 0.320 | 0.329 |
| | 336 | **0.240** | **0.280** | 0.254 | 0.286 | 0.243 | 0.280 | 0.253 | 0.289 | 0.257 | 0.305 | 0.245 | 0.282 | 0.363 | 0.366 |
| | 720 | 0.325 | 0.338 | 0.326 | 0.337 | **0.312** | **0.326** | 0.321 | 0.338 | 0.318 | 0.356 | 0.314 | 0.334 | 0.440 | 0.404 |
| | Avg | 0.224 | 0.261 | 0.236 | 0.271 | **0.222** | **0.259** | 0.233 | 0.271 | 0.239 | 0.289 | 0.225 | 0.264 | 0.329 | 0.336 |
| Traffic | 96 | **0.355** | **0.246** | 0.388 | 0.282 | 0.379 | 0.274 | 0.363 | 0.265 | 0.411 | 0.294 | 0.360 | 0.249 | 0.611 | 0.323 |
| | 192 | **0.373** | **0.253** | 0.407 | 0.290 | 0.397 | 0.282 | 0.385 | 0.273 | 0.421 | 0.298 | 0.379 | 0.256 | 0.609 | 0.327 |
| | 336 | **0.384** | **0.260** | 0.412 | 0.294 | 0.407 | 0.289 | 0.396 | 0.277 | 0.431 | 0.304 | 0.392 | 0.264 | 0.616 | 0.335 |
| | 720 | **0.427** | **0.286** | 0.450 | 0.312 | 0.440 | 0.301 | 0.445 | 0.312 | 0.468 | 0.325 | 0.432 | **0.286** | 0.656 | 0.349 |
| | Avg | **0.385** | **0.261** | 0.414 | 0.294 | 0.406 | 0.286 | 0.397 | 0.282 | 0.433 | 0.305 | 0.391 | 0.264 | 0.623 | 0.334 |
| 1$^{st}$ Count | | 29 | | 0 | | 6 | | 0 | | 0 | | 6 | | 0 | |

the Traffic and Electricity datasets, as they are included in the pre-training data for Time-MoE. The zero-shot comparison results with Time-MoE family are shown in Table 12. From the table, it is evident that, despite having significantly fewer parameters and less pre-training data, the variants of SEMPO achieve comparable forecasting performance to the large-scale Time-MoE variants across these five datasets.

**Full Results for Full-shot Experiment.** To ensure fairness and enhance the comprehensiveness of our evaluation, we have added in-distribution experiments under the standard supervised setting adopted by Time-MoE [21]. Specifically, we compare SEMPO against LLM-based time series FMs (S$^2$IP-LLM and GPT4TS) as well as strong task-specific models (iTransformer, DLinear, PatchTST and TimesNet). Results on four datasets of different scales are presented in Table 13. The results show that SEMPO consistently outperforms six advanced time series models, demonstrating that its pre-training framework effectively equips the model with strong inductive biases for downstream adaptation, despite its lightweight design.

**Full Results for Few-shot Experiment.** Table 14 and Table 15 show the 5% and 10% few-shot results for all prediction lengths on the TSLib benchmark.

**Full Results for Ablation Study.** Table 16 presents a comprehensive analysis of the impact of EASD and MoPFormer within SEMPO. As shown in the table, Group A replaces the dual-branch spectral masking in EASD with multi-band masking and random patch masking. Group B replaces MoP with Sparse MoE and prefix tuning, where sparse MoE is designed in two versions: 8 experts with activated and 3 experts with 1 activated. EASD, by performing frequency decomposition across two energy branches, helps SEMPO capture low-energy components that are easily overlooked by existing methods such as the alternative methods, leading to more accurate forecasts. The incorporation of the mixture of prompts significantly benefits SEMPO by effectively and efficiently decoupling the weights across diverse domains.

Table 14: Full few-shot results on the TSLib benchmark [25] with 5% training data. MSE and MAE are averaged over forecasting horizons $H \in \{96, 192, 336, 720\}$, where lower values indicate better prediction. **Red**: the best, **Blue**: the second best.

| Models | | SEMPO | | TTM | | Time-LLM | | GPT4TS | | S$^2$IP-LLM | | iTransformer | | DLinear | | PatchTST | | TimesNet | | Stationary | | FEDformer | | Autoformer | |
|---|---|---|---|---|---|---|---|---|---|---|---|---|---|---|---|---|---|---|---|---|---|---|---|---|---|
| Metrics | | MSE | MAE | MSE | MAE | MSE | MAE | MSE | MAE | MSE | MAE | MSE | MAE | MSE | MAE | MSE | MAE | MSE | MAE | MSE | MAE | MSE | MAE | MSE | MAE |
| ETTh1 | 96 | 0.383 | 0.408 | 0.362 | 0.389 | 0.483 | 0.464 | 0.543 | 0.506 | 0.475 | 0.458 | 0.808 | 0.610 | 0.547 | 0.503 | 0.557 | 0.519 | 0.892 | 0.625 | 0.952 | 0.650 | 0.593 | 0.529 | 0.681 | 0.570 |
| | 192 | 0.409 | 0.424 | 0.386 | 0.408 | 0.629 | 0.540 | 0.748 | 0.580 | 0.693 | 0.562 | 0.928 | 0.658 | 0.720 | 0.604 | 0.711 | 0.570 | 0.940 | 0.665 | 0.943 | 0.645 | 0.652 | 0.563 | 0.725 | 0.602 |
| | 336 | 0.425 | 0.439 | 0.400 | 0.421 | 0.768 | 0.626 | 0.754 | 0.595 | 0.760 | 0.618 | 1.475 | 0.861 | 0.984 | 0.727 | 0.816 | 0.619 | 0.945 | 0.653 | 0.935 | 0.644 | 0.731 | 0.594 | 0.761 | 0.624 |
| | 720 | - | - | - | - | - | - | - | - | - | - | - | - | - | - | - | - | - | - | - | - | - | - | - | - |
| | Avg | 0.406 | 0.423 | 0.382 | 0.405 | 0.627 | 0.543 | 0.681 | 0.560 | 0.642 | 0.546 | 1.070 | 0.710 | 0.750 | 0.611 | 0.694 | 0.569 | 0.925 | 0.647 | 0.943 | 0.646 | 0.658 | 0.562 | 0.722 | 0.598 |
| ETTh2 | 96 | 0.272 | 0.338 | 0.272 | 0.331 | 0.336 | 0.397 | 0.376 | 0.421 | 0.323 | 0.385 | 0.397 | 0.427 | 0.442 | 0.456 | 0.401 | 0.421 | 0.409 | 0.420 | 0.408 | 0.423 | 0.390 | 0.424 | 0.428 | 0.468 |
| | 192 | 0.333 | 0.378 | 0.346 | 0.383 | 0.406 | 0.425 | 0.418 | 0.441 | 0.403 | 0.420 | 0.438 | 0.445 | 0.617 | 0.542 | 0.452 | 0.455 | 0.483 | 0.464 | 0.497 | 0.468 | 0.457 | 0.465 | 0.496 | 0.504 |
| | 336 | 0.355 | 0.402 | 0.383 | 0.415 | 0.405 | 0.432 | 0.408 | 0.439 | 0.415 | 0.440 | 0.631 | 0.553 | 1.424 | 0.849 | 0.464 | 0.469 | 0.499 | 0.479 | 0.507 | 0.481 | 0.477 | 0.483 | 0.486 | 0.496 |
| | 720 | - | - | - | - | - | - | - | - | - | - | - | - | - | - | - | - | - | - | - | - | - | - | - | - |
| | Avg | 0.320 | 0.372 | 0.333 | 0.376 | 0.382 | 0.418 | 0.400 | 0.433 | 0.380 | 0.415 | 0.488 | 0.475 | 0.694 | 0.577 | 0.827 | 0.615 | 0.439 | 0.448 | 0.470 | 0.489 | 0.463 | 0.454 | 0.441 | 0.457 |
| ETTm1 | 96 | 0.307 | 0.354 | 0.341 | 0.359 | 0.316 | 0.377 | 0.386 | 0.405 | 0.303 | 0.362 | 0.589 | 0.510 | 0.332 | 0.374 | 0.399 | 0.414 | 0.606 | 0.518 | 0.823 | 0.587 | 0.628 | 0.544 | 0.726 | 0.578 |
| | 192 | 0.344 | 0.375 | 0.384 | 0.380 | 0.450 | 0.464 | 0.440 | 0.438 | 0.437 | 0.430 | 0.703 | 0.565 | 0.358 | 0.390 | 0.441 | 0.436 | 0.681 | 0.539 | 0.844 | 0.591 | 0.666 | 0.566 | 0.750 | 0.591 |
| | 336 | 0.376 | 0.393 | 0.389 | 0.396 | 0.450 | 0.424 | 0.485 | 0.459 | 0.445 | 0.423 | 0.898 | 0.641 | 0.402 | 0.416 | 0.499 | 0.467 | 0.786 | 0.597 | 0.870 | 0.603 | 0.807 | 0.628 | 0.851 | 0.659 |
| | 720 | 0.427 | 0.421 | 0.442 | 0.423 | 0.483 | 0.471 | 0.577 | 0.499 | 0.479 | 0.469 | 0.948 | 0.671 | 0.511 | 0.489 | 0.767 | 0.587 | 0.796 | 0.593 | 0.893 | 0.611 | 0.822 | 0.633 | 0.857 | 0.655 |
| | Avg | 0.363 | 0.385 | 0.389 | 0.389 | 0.425 | 0.434 | 0.472 | 0.450 | 0.416 | 0.421 | 0.784 | 0.597 | 0.400 | 0.417 | 0.526 | 0.476 | 0.717 | 0.561 | 0.857 | 0.598 | 0.730 | 0.592 | 0.796 | 0.620 |
| ETTm2 | 96 | 0.167 | 0.258 | 0.176 | 0.259 | 0.174 | 0.261 | 0.199 | 0.280 | 0.170 | 0.255 | 0.265 | 0.339 | 0.236 | 0.326 | 0.206 | 0.288 | 0.220 | 0.299 | 0.238 | 0.316 | 0.229 | 0.320 | 0.232 | 0.322 |
| | 192 | 0.225 | 0.295 | 0.244 | 0.305 | 0.215 | 0.287 | 0.256 | 0.316 | 0.228 | 0.297 | 0.310 | 0.362 | 0.306 | 0.373 | 0.264 | 0.324 | 0.311 | 0.361 | 0.298 | 0.349 | 0.394 | 0.361 | 0.291 | 0.357 |
| | 336 | 0.274 | 0.328 | 0.319 | 0.347 | 0.273 | 0.330 | 0.318 | 0.353 | 0.294 | 0.347 | 0.373 | 0.399 | 0.380 | 0.423 | 0.334 | 0.367 | 0.338 | 0.366 | 0.353 | 0.380 | 0.378 | 0.427 | 0.478 | 0.517 |
| | 720 | 0.359 | 0.381 | 0.402 | 0.403 | 0.433 | 0.412 | 0.460 | 0.436 | 0.425 | 0.404 | 0.478 | 0.454 | 0.674 | 0.583 | 0.454 | 0.432 | 0.509 | 0.465 | 0.475 | 0.445 | 0.523 | 0.510 | 0.553 | 0.538 |
| | Avg | 0.256 | 0.315 | 0.285 | 0.328 | 0.274 | 0.323 | 0.308 | 0.346 | 0.279 | 0.325 | 0.356 | 0.388 | 0.399 | 0.426 | 0.314 | 0.352 | 0.344 | 0.372 | 0.341 | 0.372 | 0.381 | 0.404 | 0.388 | 0.433 |
| Weather | 96 | 0.154 | 0.205 | 0.155 | 0.203 | 0.172 | 0.263 | 0.175 | 0.230 | 0.167 | 0.218 | 0.264 | 0.307 | 0.184 | 0.242 | 0.171 | 0.224 | 0.207 | 0.253 | 0.215 | 0.252 | 0.229 | 0.309 | 0.227 | 0.299 |
| | 192 | 0.198 | 0.246 | 0.206 | 0.257 | 0.224 | 0.271 | 0.227 | 0.276 | 0.225 | 0.273 | 0.284 | 0.326 | 0.228 | 0.283 | 0.230 | 0.277 | 0.272 | 0.307 | 0.290 | 0.307 | 0.265 | 0.317 | 0.278 | 0.333 |
| | 336 | 0.249 | 0.285 | 0.255 | 0.292 | 0.282 | 0.321 | 0.286 | 0.322 | 0.280 | 0.320 | 0.323 | 0.349 | 0.279 | 0.322 | 0.294 | 0.326 | 0.313 | 0.328 | 0.353 | 0.348 | 0.353 | 0.392 | 0.351 | 0.393 |
| | 720 | 0.321 | 0.337 | 0.330 | 0.343 | 0.366 | 0.381 | 0.366 | 0.379 | 0.359 | 0.371 | 0.366 | 0.375 | 0.364 | 0.388 | 0.384 | 0.387 | 0.400 | 0.385 | 0.452 | 0.407 | 0.391 | 0.394 | 0.387 | 0.389 |
| | Avg | 0.230 | 0.268 | 0.236 | 0.273 | 0.260 | 0.309 | 0.263 | 0.301 | 0.257 | 0.295 | 0.309 | 0.339 | 0.263 | 0.308 | 0.269 | 0.303 | 0.298 | 0.318 | 0.327 | 0.328 | 0.309 | 0.353 | 0.310 | 0.353 |
| Electricity | 96 | 0.136 | 0.234 | 0.146 | 0.241 | 0.147 | 0.242 | 0.143 | 0.241 | 0.153 | 0.251 | 0.162 | 0.264 | 0.150 | 0.251 | 0.145 | 0.244 | 0.315 | 0.389 | 0.484 | 0.518 | 0.235 | 0.322 | 0.297 | 0.367 |
| | 192 | 0.153 | 0.249 | 0.166 | 0.262 | 0.158 | 0.241 | 0.159 | 0.255 | 0.169 | 0.268 | 0.180 | 0.278 | 0.163 | 0.263 | 0.163 | 0.260 | 0.318 | 0.396 | 0.501 | 0.531 | 0.247 | 0.341 | 0.308 | 0.375 |
| | 336 | 0.168 | 0.267 | 0.185 | 0.282 | 0.178 | 0.277 | 0.179 | 0.274 | 0.183 | 0.281 | 0.207 | 0.305 | 0.175 | 0.278 | 0.183 | 0.281 | 0.340 | 0.415 | 0.574 | 0.578 | 0.267 | 0.356 | 0.354 | 0.411 |
| | 720 | 0.205 | 0.303 | 0.236 | 0.320 | 0.224 | 0.312 | 0.233 | 0.323 | 0.239 | 0.324 | 0.258 | 0.339 | 0.219 | 0.311 | 0.233 | 0.323 | 0.635 | 0.613 | 0.952 | 0.786 | 0.318 | 0.394 | 0.426 | 0.466 |
| | Avg | 0.165 | 0.263 | 0.183 | 0.276 | 0.179 | 0.268 | 0.178 | 0.273 | 0.186 | 0.281 | 0.201 | 0.296 | 0.176 | 0.275 | 0.181 | 0.277 | 0.402 | 0.453 | 0.627 | 0.603 | 0.266 | 0.353 | 0.346 | 0.404 |
| Traffic | 96 | 0.399 | 0.283 | 0.411 | 0.293 | 0.414 | 0.291 | 0.419 | 0.298 | 0.410 | 0.289 | 0.431 | 0.312 | 0.427 | 0.304 | 0.404 | 0.286 | 0.854 | 0.492 | 1.468 | 0.821 | 0.670 | 0.421 | 0.795 | 0.481 |
| | 192 | 0.411 | 0.287 | 0.425 | 0.300 | 0.419 | 0.291 | 0.434 | 0.305 | 0.415 | 0.295 | 0.456 | 0.326 | 0.447 | 0.315 | 0.412 | 0.294 | 0.894 | 0.517 | 1.509 | 0.838 | 0.653 | 0.405 | 0.837 | 0.503 |
| | 336 | 0.421 | 0.292 | 0.446 | 0.315 | 0.437 | 0.314 | 0.449 | 0.313 | 0.433 | 0.310 | 0.465 | 0.334 | 0.478 | 0.333 | 0.439 | 0.310 | 0.853 | 0.471 | 1.602 | 0.860 | 0.707 | 0.445 | 0.867 | 0.523 |
| | 720 | - | - | - | - | - | - | - | - | - | - | - | - | - | - | - | - | - | - | - | - | - | - | - | - |
| | Avg | 0.410 | 0.287 | 0.427 | 0.303 | 0.423 | 0.298 | 0.434 | 0.305 | 0.419 | 0.298 | 0.450 | 0.324 | 0.450 | 0.317 | 0.418 | 0.296 | 0.867 | 0.493 | 1.526 | 0.839 | 0.676 | 0.423 | 0.833 | 0.502 |
| $1^{st}$ Count | | 48 | | 11 | | 4 | | 0 | | 2 | | 0 | | 0 | | 0 | | 0 | | 0 | | 0 | | 0 | |

Table 15: Full few-shot results on the TSLib benchmark [25] with 10% training data. MSE and MAE are averaged over forecasting horizons $H \in \{96, 192, 336, 720\}$, where lower values indicate better prediction. **Red**: the best, **Blue**: the second best.

| Models | | SEMPO | | TTM | | Time-LLM | | GPT4TS | | S²P-LLM | | iTransformer | | DLinear | | PatchTST | | TimesNet | | Stationary | | FEDformer | | Autoformer | |
|---|---|---|---|---|---|---|---|---|---|---|---|---|---|---|---|---|---|---|---|---|---|---|---|---|---|
| Metrics | | MSE | MAE | MSE | MAE | MSE | MAE | MSE | MAE | MSE | MAE | MSE | MAE | MSE | MAE | MSE | MAE | MSE | MAE | MSE | MAE | MSE | MAE | MSE | MAE |
| ETTh1 | 96 | 0.378 | 0.405 | 0.362 | 0.389 | 0.448 | 0.460 | 0.458 | 0.456 | 0.463 | 0.459 | 0.790 | 0.586 | 0.492 | 0.495 | 0.516 | 0.485 | 0.861 | 0.628 | 0.918 | 0.639 | 0.512 | 0.499 | 0.613 | 0.552 |
| | 192 | 0.407 | 0.424 | 0.386 | 0.408 | 0.484 | 0.483 | 0.570 | 0.516 | 0.482 | 0.487 | 0.837 | 0.609 | 0.565 | 0.538 | 0.598 | 0.524 | 0.797 | 0.593 | 0.915 | 0.629 | 0.624 | 0.555 | 0.722 | 0.598 |
| | 336 | 0.426 | 0.439 | 0.399 | 0.420 | 0.589 | 0.540 | 0.608 | 0.535 | 0.603 | 0.543 | 0.780 | 0.575 | 0.721 | 0.622 | 0.657 | 0.550 | 0.941 | 0.648 | 0.939 | 0.644 | 0.691 | 0.574 | 0.750 | 0.619 |
| | 720 | 0.456 | 0.474 | 0.465 | 0.477 | 0.700 | 0.604 | 0.725 | 0.591 | 0.713 | 0.588 | 1.234 | 0.811 | 0.986 | 0.743 | 0.762 | 0.610 | 0.877 | 0.641 | 0.887 | 0.645 | 0.728 | 0.614 | 0.721 | 0.616 |
| | Avg | 0.417 | 0.435 | 0.403 | 0.423 | 0.556 | 0.522 | 0.590 | 0.525 | 0.565 | 0.524 | 0.910 | 0.860 | 0.691 | 0.600 | 0.633 | 0.542 | 0.869 | 0.628 | 0.915 | 0.639 | 0.639 | 0.561 | 0.702 | 0.596 |
| ETTh2 | 96 | 0.276 | 0.340 | 0.275 | 0.335 | 0.275 | 0.326 | 0.331 | 0.374 | 0.300 | 0.360 | 0.404 | 0.435 | 0.357 | 0.411 | 0.353 | 0.389 | 0.378 | 0.409 | 0.389 | 0.411 | 0.382 | 0.416 | 0.413 | 0.451 |
| | 192 | 0.332 | 0.376 | 0.345 | 0.383 | 0.374 | 0.373 | 0.402 | 0.411 | 0.372 | 0.371 | 0.470 | 0.474 | 0.569 | 0.519 | 0.403 | 0.414 | 0.490 | 0.467 | 0.473 | 0.455 | 0.478 | 0.474 | 0.474 | 0.477 |
| | 336 | 0.354 | 0.399 | 0.384 | 0.416 | 0.406 | 0.429 | 0.406 | 0.433 | 0.389 | 0.413 | 0.489 | 0.485 | 0.671 | 0.572 | 0.426 | 0.441 | 0.537 | 0.494 | 0.507 | 0.480 | 0.504 | 0.501 | 0.547 | 0.543 |
| | 720 | 0.395 | 0.433 | 0.419 | 0.450 | 0.427 | 0.449 | 0.449 | 0.464 | 0.403 | 0.426 | 0.593 | 0.538 | 0.824 | 0.648 | 0.477 | 0.480 | 0.510 | 0.491 | 0.477 | 0.472 | 0.499 | 0.509 | 0.516 | 0.523 |
| | Avg | 0.339 | 0.387 | 0.355 | 0.391 | 0.370 | 0.394 | 0.397 | 0.421 | 0.366 | 0.392 | 0.489 | 0.483 | 0.605 | 0.538 | 0.415 | 0.431 | 0.479 | 0.465 | 0.462 | 0.455 | 0.466 | 0.475 | 0.488 | 0.499 |
| ETTm1 | 96 | 0.304 | 0.355 | 0.346 | 0.363 | 0.346 | 0.388 | 0.390 | 0.404 | 0.353 | 0.390 | 0.709 | 0.556 | 0.352 | 0.392 | 0.410 | 0.419 | 0.583 | 0.501 | 0.761 | 0.568 | 0.578 | 0.518 | 0.774 | 0.614 |
| | 192 | 0.340 | 0.375 | 0.375 | 0.395 | 0.373 | 0.416 | 0.429 | 0.423 | 0.368 | 0.403 | 0.717 | 0.548 | 0.382 | 0.412 | 0.437 | 0.434 | 0.630 | 0.528 | 0.781 | 0.574 | 0.617 | 0.546 | 0.754 | 0.592 |
| | 336 | 0.372 | 0.392 | 0.398 | 0.392 | 0.413 | 0.426 | 0.469 | 0.439 | 0.417 | 0.428 | 0.735 | 0.575 | 0.419 | 0.434 | 0.476 | 0.454 | 0.725 | 0.568 | 0.803 | 0.587 | 0.998 | 0.775 | 0.869 | 0.677 |
| | 720 | 0.427 | 0.423 | 0.443 | 0.429 | 0.485 | 0.476 | 0.569 | 0.498 | 0.473 | 0.468 | 0.752 | 0.584 | 0.490 | 0.477 | 0.681 | 0.556 | 0.769 | 0.549 | 0.844 | 0.581 | 0.693 | 0.579 | 0.810 | 0.630 |
| | Avg | 0.360 | 0.386 | 0.390 | 0.394 | 0.404 | 0.427 | 0.464 | 0.441 | 0.402 | 0.422 | 0.728 | 0.565 | 0.411 | 0.429 | 0.501 | 0.466 | 0.677 | 0.537 | 0.797 | 0.578 | 0.722 | 0.605 | 0.802 | 0.628 |
| ETTm2 | 96 | 0.166 | 0.256 | 0.176 | 0.260 | 0.177 | 0.261 | 0.188 | 0.269 | 0.140 | 0.242 | 0.245 | 0.322 | 0.213 | 0.303 | 0.191 | 0.274 | 0.212 | 0.285 | 0.229 | 0.308 | 0.291 | 0.399 | 0.352 | 0.454 |
| | 192 | 0.223 | 0.295 | 0.242 | 0.304 | 0.241 | 0.314 | 0.251 | 0.309 | 0.207 | 0.293 | 0.274 | 0.338 | 0.278 | 0.345 | 0.252 | 0.317 | 0.270 | 0.323 | 0.291 | 0.343 | 0.307 | 0.379 | 0.694 | 0.593 |
| | 336 | 0.276 | 0.329 | 0.315 | 0.345 | 0.274 | 0.327 | 0.307 | 0.346 | 0.264 | 0.331 | 0.361 | 0.394 | 0.338 | 0.385 | 0.306 | 0.353 | 0.323 | 0.353 | 0.348 | 0.376 | 0.543 | 0.559 | 2.408 | 1.407 |
| | 720 | 0.357 | 0.381 | 0.394 | 0.399 | 0.417 | 0.390 | 0.426 | 0.417 | 0.381 | 0.387 | 0.467 | 0.442 | 0.436 | 0.440 | 0.433 | 0.427 | 0.474 | 0.449 | 0.461 | 0.438 | 0.712 | 0.614 | 1.913 | 1.166 |
| | Avg | 0.255 | 0.315 | 0.281 | 0.327 | 0.277 | 0.323 | 0.293 | 0.335 | 0.248 | 0.313 | 0.336 | 0.373 | 0.316 | 0.368 | 0.296 | 0.343 | 0.320 | 0.353 | 0.332 | 0.366 | 0.463 | 0.488 | 1.342 | 0.930 |
| Weather | 96 | 0.152 | 0.204 | 0.152 | 0.199 | 0.161 | 0.210 | 0.163 | 0.215 | 0.154 | 0.201 | 0.253 | 0.307 | 0.171 | 0.224 | 0.165 | 0.215 | 0.184 | 0.230 | 0.192 | 0.234 | 0.188 | 0.253 | 0.221 | 0.297 |
| | 192 | 0.196 | 0.243 | 0.193 | 0.245 | 0.204 | 0.248 | 0.210 | 0.254 | 0.195 | 0.241 | 0.292 | 0.328 | 0.215 | 0.263 | 0.210 | 0.257 | 0.245 | 0.283 | 0.269 | 0.295 | 0.250 | 0.304 | 0.270 | 0.322 |
| | 336 | 0.246 | 0.282 | 0.246 | 0.282 | 0.261 | 0.302 | 0.256 | 0.292 | 0.260 | 0.302 | 0.322 | 0.346 | 0.258 | 0.299 | 0.259 | 0.297 | 0.305 | 0.321 | 0.370 | 0.357 | 0.312 | 0.346 | 0.320 | 0.351 |
| | 720 | 0.319 | 0.336 | 0.336 | 0.346 | 0.309 | 0.332 | 0.321 | 0.339 | 0.303 | 0.329 | 0.365 | 0.374 | 0.320 | 0.346 | 0.332 | 0.346 | 0.381 | 0.371 | 0.441 | 0.405 | 0.387 | 0.393 | 0.390 | 0.396 |
| | Avg | 0.228 | 0.266 | 0.232 | 0.268 | 0.234 | 0.273 | 0.238 | 0.275 | 0.228 | 0.268 | 0.308 | 0.338 | 0.241 | 0.283 | 0.242 | 0.279 | 0.279 | 0.301 | 0.318 | 0.323 | 0.284 | 0.324 | 0.300 | 0.342 |
| Electricity | 96 | 0.134 | 0.231 | 0.140 | 0.237 | 0.139 | 0.241 | 0.139 | 0.237 | 0.142 | 0.243 | 0.154 | 0.257 | 0.150 | 0.253 | 0.140 | 0.238 | 0.299 | 0.373 | 0.420 | 0.466 | 0.231 | 0.323 | 0.261 | 0.348 |
| | 192 | 0.151 | 0.247 | 0.163 | 0.259 | 0.151 | 0.248 | 0.156 | 0.252 | 0.163 | 0.260 | 0.171 | 0.272 | 0.164 | 0.264 | 0.160 | 0.255 | 0.305 | 0.379 | 0.411 | 0.459 | 0.261 | 0.356 | 0.338 | 0.406 |
| | 336 | 0.167 | 0.267 | 0.180 | 0.275 | 0.169 | 0.270 | 0.175 | 0.270 | 0.173 | 0.270 | 0.196 | 0.295 | 0.181 | 0.282 | 0.180 | 0.276 | 0.319 | 0.391 | 0.434 | 0.473 | 0.360 | 0.445 | 0.410 | 0.474 |
| | 720 | 0.201 | 0.298 | 0.241 | 0.326 | 0.240 | 0.322 | 0.233 | 0.317 | 0.237 | 0.320 | 0.263 | 0.348 | 0.223 | 0.321 | 0.241 | 0.323 | 0.369 | 0.426 | 0.510 | 0.521 | 0.530 | 0.585 | 0.715 | 0.685 |
| | Avg | 0.163 | 0.261 | 0.181 | 0.274 | 0.175 | 0.270 | 0.176 | 0.269 | 0.178 | 0.273 | 0.196 | 0.293 | 0.180 | 0.280 | 0.180 | 0.273 | 0.323 | 0.392 | 0.444 | 0.480 | 0.346 | 0.427 | 0.431 | 0.478 |
| Traffic | 96 | 0.395 | 0.277 | 0.408 | 0.292 | 0.418 | 0.291 | 0.414 | 0.297 | 0.401 | 0.285 | 0.448 | 0.329 | 0.419 | 0.298 | 0.403 | 0.289 | 0.719 | 0.416 | 1.412 | 0.802 | 0.639 | 0.400 | 0.672 | 0.405 |
| | 192 | 0.407 | 0.282 | 0.417 | 0.295 | 0.414 | 0.296 | 0.426 | 0.301 | 0.410 | 0.293 | 0.487 | 0.360 | 0.434 | 0.305 | 0.415 | 0.296 | 0.748 | 0.428 | 1.419 | 0.806 | 0.637 | 0.416 | 0.727 | 0.424 |
| | 336 | 0.417 | 0.287 | 0.430 | 0.302 | 0.421 | 0.311 | 0.434 | 0.303 | 0.425 | 0.314 | 0.514 | 0.372 | 0.449 | 0.313 | 0.426 | 0.304 | 0.853 | 0.471 | 1.443 | 0.815 | 0.655 | 0.427 | 0.749 | 0.454 |
| | 720 | 0.450 | 0.304 | 0.476 | 0.331 | 0.462 | 0.327 | 0.487 | 0.337 | 0.470 | 0.330 | 0.532 | 0.383 | 0.484 | 0.336 | 0.474 | 0.331 | 1.485 | 0.825 | 1.539 | 0.837 | 0.722 | 0.456 | 0.847 | 0.499 |
| | Avg | 0.417 | 0.287 | 0.432 | 0.305 | 0.429 | 0.306 | 0.440 | 0.310 | 0.426 | 0.305 | 0.495 | 0.361 | 0.447 | 0.313 | 0.430 | 0.305 | 0.951 | 0.535 | 1.453 | 0.815 | 0.663 | 0.425 | 0.749 | 0.446 |
| 1ˢᵗ Count | | 45 | | 15 | | 4 | | 0 | | 13 | | 0 | | 0 | | 0 | | 0 | | 0 | | 0 | | 0 | |

Table 16: Full results for our ablation studies. MSE and MAE for zero-shot forecasting on the TSLib benchmark [25], evaluated with different model components. Group A replaces dual-branch spectral masking with alternative masking methods, while Group B replaces MoP with alternative specialization methods.

| Datasets | | ETTh1 | | ETTh2 | | ETTm1 | | ETTm2 | | Weather | | Electricity | | Traffic | |
|---|---|---|---|---|---|---|---|---|---|---|---|---|---|---|---|---|
| Metrics | | MSE | MAE | MSE | MAE | MSE | MAE | MSE | MAE | MSE | MAE | MSE | MAE | MSE | MAE |
| SEMPO | 96 | 0.384 | 0.408 | 0.282 | 0.342 | 0.466 | 0.443 | 0.196 | 0.286 | 0.171 | 0.228 | 0.168 | 0.271 | 0.441 | 0.333 |
| | 192 | 0.409 | 0.426 | 0.334 | 0.384 | 0.484 | 0.455 | 0.252 | 0.323 | 0.218 | 0.269 | 0.183 | 0.283 | 0.456 | 0.339 |
| | 336 | 0.417 | 0.433 | 0.355 | 0.403 | 0.506 | 0.469 | 0.306 | 0.354 | 0.267 | 0.304 | 0.198 | 0.297 | 0.467 | 0.344 |
| | 720 | 0.432 | 0.454 | 0.395 | 0.435 | 0.557 | 0.498 | 0.391 | 0.404 | 0.336 | 0.350 | 0.238 | 0.329 | 0.503 | 0.360 |
| | avg | 0.410 | 0.430 | 0.341 | 0.391 | 0.503 | 0.466 | 0.286 | 0.341 | 0.248 | 0.287 | 0.196 | 0.295 | 0.466 | 0.344 |
| **A.1** Multi-band Masking | 96 | 0.438 | 0.441 | 0.348 | 0.389 | 0.533 | 0.476 | 0.239 | 0.317 | 0.183 | 0.248 | 0.180 | 0.284 | 0.518 | 0.348 |
| | 192 | 0.455 | 0.453 | 0.403 | 0.421 | 0.538 | 0.480 | 0.286 | 0.343 | 0.231 | 0.286 | 0.192 | 0.295 | 0.524 | 0.351 |
| | 336 | 0.463 | 0.461 | 0.424 | 0.438 | 0.558 | 0.491 | 0.338 | 0.376 | 0.279 | 0.324 | 0.205 | 0.308 | 0.535 | 0.355 |
| | 720 | 0.495 | 0.492 | 0.519 | 0.498 | 0.620 | 0.527 | 0.506 | 0.471 | 0.352 | 0.377 | 0.240 | 0.338 | 0.572 | 0.372 |
| | avg | 0.462 | 0.461 | 0.423 | 0.436 | 0.562 | 0.493 | 0.342 | 0.376 | 0.261 | 0.308 | 0.204 | 0.306 | 0.537 | 0.356 |
| **A.3** Random Patch Masking | 96 | 0.422 | 0.437 | 0.327 | 0.381 | 0.546 | 0.480 | 0.246 | 0.326 | 0.189 | 0.256 | 0.218 | 0.324 | 0.625 | 0.397 |
| | 192 | 0.440 | 0.450 | 0.385 | 0.416 | 0.553 | 0.487 | 0.302 | 0.359 | 0.235 | 0.295 | 0.231 | 0.336 | 0.631 | 0.399 |
| | 336 | 0.449 | 0.461 | 0.417 | 0.438 | 0.573 | 0.498 | 0.351 | 0.388 | 0.280 | 0.328 | 0.247 | 0.349 | 0.648 | 0.404 |
| | 720 | 0.476 | 0.492 | 0.471 | 0.476 | 0.625 | 0.526 | 0.461 | 0.449 | 0.343 | 0.375 | 0.278 | 0.373 | 0.686 | 0.418 |
| | avg | 0.446 | 0.460 | 0.400 | 0.428 | 0.574 | 0.498 | 0.340 | 0.381 | 0.261 | 0.313 | 0.243 | 0.345 | 0.647 | 0.404 |
| **B.1** Sparse MoE (8 experts, 2 activated) | 96 | 0.648 | 0.532 | 0.354 | 0.405 | 0.610 | 0.509 | 0.246 | 0.340 | 0.185 | 0.248 | 0.211 | 0.316 | 0.600 | 0.394 |
| | 192 | 0.689 | 0.551 | 0.416 | 0.439 | 0.598 | 0.509 | 0.291 | 0.364 | 0.235 | 0.289 | 0.226 | 0.328 | 0.618 | 0.400 |
| | 336 | 0.692 | 0.558 | 0.448 | 0.461 | 0.610 | 0.518 | 0.350 | 0.398 | 0.283 | 0.326 | 0.241 | 0.342 | 0.626 | 0.401 |
| | 720 | 0.720 | 0.583 | 0.521 | 0.509 | 0.659 | 0.545 | 0.491 | 0.477 | 0.350 | 0.376 | 0.274 | 0.369 | 0.664 | 0.416 |
| | avg | 0.687 | 0.556 | 0.434 | 0.453 | 0.619 | 0.520 | 0.344 | 0.394 | 0.263 | 0.309 | 0.238 | 0.338 | 0.627 | 0.402 |
| **B.1** Sparse MoE (3 experts, 1 activated) | 96 | 0.419 | 0.434 | 0.295 | 0.358 | 0.482 | 0.456 | 0.220 | 0.306 | 0.178 | 0.236 | 0.195 | 0.299 | 0.506 | 0.380 |
| | 192 | 0.435 | 0.445 | 0.356 | 0.396 | 0.498 | 0.479 | 0.277 | 0.343 | 0.225 | 0.274 | 0.211 | 0.312 | 0.522 | 0.388 |
| | 336 | 0.444 | 0.453 | 0.378 | 0.414 | 0.514 | 0.485 | 0.327 | 0.372 | 0.273 | 0.308 | 0.224 | 0.322 | 0.531 | 0.390 |
| | 720 | 0.468 | 0.478 | 0.404 | 0.441 | 0.566 | 0.520 | 0.410 | 0.422 | 0.339 | 0.353 | 0.263 | 0.352 | 0.569 | 0.406 |
| | avg | 0.441 | 0.452 | 0.358 | 0.402 | 0.515 | 0.485 | 0.308 | 0.360 | 0.253 | 0.292 | 0.223 | 0.321 | 0.532 | 0.391 |
| **B.2** Prefix Tuning | 96 | 0.402 | 0.426 | 0.298 | 0.357 | 0.486 | 0.457 | 0.220 | 0.307 | 0.192 | 0.248 | 0.189 | 0.291 | 0.468 | 0.354 |
| | 192 | 0.425 | 0.441 | 0.351 | 0.394 | 0.491 | 0.462 | 0.277 | 0.342 | 0.240 | 0.289 | 0.203 | 0.304 | 0.485 | 0.362 |
| | 336 | 0.437 | 0.451 | 0.389 | 0.431 | 0.513 | 0.476 | 0.328 | 0.373 | 0.286 | 0.322 | 0.218 | 0.318 | 0.495 | 0.365 |
| | 720 | 0.456 | 0.476 | 0.398 | 0.437 | 0.565 | 0.507 | 0.413 | 0.422 | 0.356 | 0.368 | 0.258 | 0.349 | 0.529 | 0.379 |
| | avg | 0.430 | 0.448 | 0.359 | 0.404 | 0.513 | 0.475 | 0.309 | 0.361 | 0.268 | 0.306 | 0.217 | 0.315 | 0.494 | 0.365 |

