# OpenReview forum: "SEMPO: Lightweight Foundation Models for Time Series Forecasting"
_NeurIPS.cc/2025/Conference — NeurIPS 2025 poster_

### Official Review · Reviewer_EgfG · 2025-06-05

**Clarity:** 3
**Significance:** 3
**Originality:** 3
**Rating:** 5
**Confidence:** 4

**Summary:**

The paper proposes a lightweight foundation model for time series forecasting, SEMPO, which includes an Energy-Aware Spectral Decomposition (EASD) module and a Mixture-of-Prompts enabled Transformer (MoPFormer) module. The former leverages high- and low-energy frequency signals to improve data utilization, while the latter achieves parameter-efficient adaptation through a lightweight prompt-based expert system. Experiments show that SEMPO, with 6.5 million parameters and pre-trained on 83 million data samples, outperforms mainstream models in terms of prediction accuracy under zero-shot and few-shot scenarios, demonstrating its efficiency and generalization capability.

**Questions:**

The authors can refer to the weakness listed above.

**Ethical Concerns:**

["NO or VERY MINOR ethics concerns only"]

**Final Justification:**

The paper proposes a lightweight foundation model for time series forecasting, SEMPO, which includes an Energy-Aware Spectral Decomposition (EASD) module and a Mixture-of-Prompts enabled Transformer (MoPFormer) module. The former leverages high- and low-energy frequency signals to improve data utilization, while the latter achieves parameter-efficient adaptation through a lightweight prompt-based expert system. Experiments show that SEMPO, with 6.5 million parameters and pre-trained on 83 million data samples, outperforms mainstream models in terms of prediction accuracy under zero-shot and few-shot scenarios, demonstrating its efficiency and generalization capability. The author's response has largely addressed my concerns. Considering the significance of SEMPO for lightweight time series forecasting foundation models, I have raised my score to 5.

**Limitations:**

The authors can refer to the weakness listed above.

**Paper Formatting Concerns:**

The authors can refer to the weakness listed above.

**Quality:**

3

**Strengths And Weaknesses:**

**Strengths**

1. Proposes a lightweight SEMPO model that achieves zero-/few-shot prediction advantages with only 6.5 million parameters and 83 million pre-training data, offering high resource efficiency and practical value.
2. The experiments cover multiple datasets and baseline models, including ablation studies and visualization analyses, providing thorough validation.
3. The paper is well-structured, with clear formulas, standardized figures, precise technical descriptions, and smooth logical flow.

**References**

1. It is recommended to include full-shot experiments to evaluate model performance with the full dataset, thereby validating the benefits of pre-training.
2. A comparison with frequency debiasing models such as Fedformer [1] should be added to highlight the superiority of the EASD module.
3. The description of the two-stage training paradigm in the main text is too brief; more details should be provided.
4. The EASD module uses a masking mechanism for data augmentation, which actually increases the amount of training data. Therefore, further verification is needed to confirm whether the claimed reduction in pre-training cost is valid.
5. The zero-shot experiment in Table 1 and the efficiency analysis in Figure 6 do not include the lightweight model TTM. It is suggested to add its results to fully illustrate the performance gradient of the lightweight methods. Additionally, since new versions of Chronos and TimesFM are available, it is advisable to update the comparisons to the latest state-of-the-art models to ensure the conclusions remain current with field advancements.
6. The visualization of $Z_{Lec}$ in Figure 2 appears to contain an error; the complementary relationship between $Z_{Lec}$ and $Z_{Hec}$ should be corrected.
7. In Figure 4(b), the low-energy feature response of Moirai seems to reflect prediction inaccuracies rather than being completely neglected. The authors are encouraged to provide a more reasonable explanation for this phenomenon.

**Weaknesses**

[1] Fredformer: Frequency Debiased Transformer for Time Series Forecasting, KDD 2024

---

> ### Author Rebuttal · Authors · 2025-07-31
>
> Thank you very much for the constructive comments. We are grateful for the positive comments on the technical contribution, sound experiments and well-structured presentation. Please see our detailed response below.
>
> >**Weakness #1** Include full-shot experiments to evaluate model performance.
>
> Please see our detailed response to **Reviewer Y38u**’s **Weakness #1**, where we added the full-shot experiments on four representative datasets of different scales from the TSLib benchmark.
>
> >**Weakness #2** Compare with frequency debiasing model Fredformer.
>
> Thank you for highlighting the debiasing model Fredformer. Following your suggestion, we added a comparison between SEMPO and Fredformer under 10\% few-shot and full-shot settings, as summarized in **Table A1** below. SEMPO demonstrates strong capabilities, consistently outperforming Fredformer under both settings in both MSE and MAE. While Fredformer shifts attention toward mid- and high-frequency components, it overlooks energy-level bias, as low-energy signals can occur across all frequency bands. In contrast, the EASD module in SEMPO  suppresses dominant high-energy components to better capture subtle yet informative low-energy patterns, thereby reducing information loss and improving the forecasting generalization.
>
> ```
> Table A1. SEMPO vs. Fredformer with 10% and 100% training data. Results are averaged over horizons of {96,192,336,720}.
> ```
> |Dataset|Metrics|ETTh1|ETTh2|ETTTm1|ETTm2|Weather|Electricity|Traffic|
> |---|---|---|---|---|---|---|---|---|
> |SEMPO (10\%)|MSE|**0.417**|**0.339**|**0.361**|**0.256**|**0.228**|**0.163**|**0.417**|
> ||MAE|**0.436**|**0.387**|**0.386**|**0.315**|**0.266**|**0.261**|**0.288**|
> |Fredformer (10\%)|MSE|0.592|0.393|0.473|0.279|0.229|0.187|0.428|
> ||MAE|0.527|0.425|0.458|0.332|0.279|0.279|0.299|
> |SEMPO (100\%)|MSE|**0.419**|**0.340**|0.356|**0.250**|**0.224**|0.172|**0.385**|
> ||MAE|**0.432**|**0.386**|0.383|**0.313**|**0.261**|0.270|**0.261**|
> |Fredformer (100\%)|MSE|0.424|0.348|**0.350**|0.256|0.225|**0.161**|0.394|
> ||MAE|0.435|0.390|**0.379**|0.317|0.265|**0.256**|0.273|
>
> >**Weakness #3** Overly brief description of the two-stage training paradigm in the main text.
>
> We have explained the details of two-stage training paradigm in **Lines 882-902 of Appendix B.2**. For clarity, we will move this to the main text in the final version.
>
> >**Weakness #4** The claimed reduction in pre-training cost needs further verification.
>
> The pre-training data reduction we argue in the paper is mainly focused on the original data size, not considering the data size resulting from data augmentation techniques. However, even we take into account of the data size increment due to the data augmentation, our data reduction argument still holds.  Specifically, random masking is a widely adopted strategy in recent time series foundation models (FMs), such as TimesFM [Ref1] and Moment [Ref2], meaning that their pre-training data size is even larger than the one, TimesFM (100B) and Moment (1.13B), we reported in Table 1 if we consider the augmented samples as part of the pre-training data size.
>
> Unlike these FMs that merely apply masking on original samples, SEMPO duplicates each sample $N_m$ times across both energy branches, yielding a training size of $83M \times N_m \times 2$. Yet, despite this expansion, the total pre-training data of SEMPO remains substantially smaller than the billion-scale data required by other FMs such as TimesFM (100B) and Moment (1.13B). More importantly, SEMPO requires only 83M time points as original pre-training data, making it well-suited for deployment in resource-constrained environments. The cost introduced by duplication is already reflected in the reported memory usage and computational overhead.
> Therefore, our claim of reduced pre-training cost remains **valid**. We will add these discussions in the final version.
>
>
> >**Weakness #5** Lack of comparison to TTM in zero-shot and efficiency comparisons, and results of new versions of Chronos and TimesFM.
>
> To clarify, we have provided zero-shot comparison results with the lightweight **MLP-based** FM TTM in **Table 9 in Appendix D.6**, as explained in **Lines 264-267**, since **Table 1** focuses on zero-shot results against **Transformer-based** FMs.
>
> Following your suggestion, we have additionally included an efficiency comparison with TTM on ETTh1 and ETTm2, summarized in **Table A2** below. While SEMPO is less competitive in parameter size and inference time due to its more complex Transformer-based architecture, it uses only 83M pre-training data (vs. TTM's 1B) and still achieves lower MSE on both datasets. Moreover, SEMPO outperforms TTM in **31 out of 50 cases** across five diverse datasets (see **Table 9 in Appendix D.6**), with particularly substantial advantages on large-scale datasets such as Traffic and Electricity, demonstrating stronger knowledge transfer capability and superior sample efficiency.
>
> ```
> Table A2. Efficiency comparison with TTM on ETTh1 and ETTm2.
> ```
> |Datasets|Models|Inference Time (Seconds)|Parameters (M)|MSE|
> |-|-|-|-|-|
> ETTh1|TTM|3.2|1|0.411|
> ||SEMPO|22|6.5|**0.410**|
> ETTm2|TTM|5.2|1|0.288|
> ||SEMPO|67|6.5|**0.286**|
>
>
> To address your concern regarding the updated FMs, we added zero-shot results for the latest **Chronos-Bolt and TimesFM (500M)**, summarized in **Table A3** below for an up-to-date comparison. Across five datasets, Chronos-Bolt achieves the best performance in **16** cases, SEMPO in **12**, and TimesFM (500M) in **5**. We can observe that SEMPO significantly outperforms TimesFM (500M). It should be noted that, even with the small version of Chronos-Bolt (i.e., Chronos-Bolt$_S$), it is trained on nearly **100B** time points with **48M** parameters. In contrast, SEMPO is trained on only **83M** points with just **6.5M** parameters, yet it still ranks in second in terms of the overall performance. This compelling data efficiency–performance balance highlights SEMPO’s promise as an effective lightweight FM for resource-constrained environments, to which the other FMs such as Chronos-Bolt and TimesFM are infeasible.
>
> ```
> Table A3. Zero-shot comparison on five datasets with Chronos-Bolt and TimesFM (500M).
> ```
>
> |Dataset|Horizon|SEMPO||Chronos-Bolt$_S$||Chronos-Bolt$_L$||TimesFM (500M)||
> |---|---|---|---|---|---|---|---|---|---|
> |||MSE|MAE|MSE|MAE|MSE|MAE|MSE|MAE|MSE|MAE|
> |ETTh1|192|**0.409**|0.426|0.451|0.415|0.435|0.412|0.428|**0.409**|
> ||336|**0.417**|0.433|0.497|0.437|0.476|0.433|0.451|**0.424**|
> ||720|**0.432**|0.454|0.499|0.456|0.480|0.449|0.463|**0.447**|
> |ETTh2|192|**0.334**|0.384|0.351|**0.371**|0.355|**0.371**|0.370|0.385|
> ||336|**0.355**|**0.403**|0.396|0.405|0.400|**0.403**|0.416|0.420|
> ||720|**0.395**|0.435|0.413|0.429|0.414|**0.422**|0.443|0.448|
> |ETTm1|192|0.484|0.455|0.382|**0.366**|**0.381**|**0.366**|0.388|0.376|
> ||336|0.506|0.469|0.433|0.396|0.432|**0.393**|**0.426**|0.402|
> ||720|0.557|0.498|0.530|0.447|0.525|**0.439**|**0.491**|0.444|
> |ETTm2|192|0.252|0.323|**0.239**|**0.291**|0.247|0.295|0.268|0.307|
> ||336|**0.305**|0.354|0.306|**0.334**|0.314|0.336|0.327|0.347|
> ||720|**0.391**|0.404|0.413|0.398|0.415|**0.395**|0.421|0.407|
> |Weather|192|**0.216**|0.269|0.218|**0.250**|0.222|0.258|-|-|
> ||336|**0.267**|0.304|0.275|**0.290**|0.282|0.298|-|-|
> ||720|**0.336**|0.350|0.354|**0.338**|0.366|0.348|-|-|
> |**1$^{st}$ Count**||12||**16**||||5||
>
> >**Weakness #6** The visualization of $Z_{Lec}$ in Figure 2 appears to contain an error.
>
> We have carefully checked Figure 2, and confirm that $Z_{Lec}$ and $Z_{Hec}$ are indeed complementary, with $Z_{Lec}=Z-Z_{Hec}$.
>
> >**Weakness #7** In Figure 4(b), the low-energy feature response of Moirai seems to reflect prediction inaccuracies rather than being completely neglected.
>
> Yes, your observation is correct. As shown by the spectral discrepancy between the prediction and ground truth, Moirai$_L$ exhibits non-negligible responses in low-energy regions, in contrast to Chronos$_S$, which largely neglects low-energy components. However, these responses in Moirai$_L$ do not consistently align with the spectral peaks of the ground truth. This indicates that while Moirai$_L$ is able to partially capture low-energy signals, it still predominantly attends to high-energy components, resulting in less accurate learning of low-energy features, such as the important positional information. We will clarify this in the final version.
>
> >**References**:
>
> - [Ref1] A Decoder-only Foundation Model for Time-Series Forecasting, ICML 2024
>
> - [Ref2] MOMENT: A Family of Open Time-series Foundation Models, ICML 2024

---

> > ### Comment · Reviewer_EgfG · 2025-08-01
> >
> > Thanks for the your reply. This has largely addressed my concerns. Considering the significance of SEMPO for lightweight time series forecasting foundation models, I have raised my score to 5.

---

> > > ### Author Response · Authors · 2025-08-01
> > > **Response to Reviewer EgfG**
> > >
> > > Dear Reviewer EgfG,
> > >
> > > Thank you very much for your appreciation of SEMPO and for raising your recommendation score. We're very pleased that our response has largely addressed your concerns. Please kindly let us know if there are any further questions. Thanks again！
> > >
> > > Best regards,
> > >
> > > Authors

---

### Official Review · Reviewer_uD87 · 2025-06-26

**Clarity:** 3
**Significance:** 3
**Originality:** 3
**Rating:** 5
**Confidence:** 4

**Summary:**

The paper introduces SEMPO, a lightweight foundation model for general-purpose time series forecasting that achieves strong performance while using fewer parameters and less pre-training data than existing large-scale models. SEMPO integrates two key features: (1) an Energy-Aware Spectral Decomposition (EASD) module that transforms time series into the frequency domain and models both high- and low-energy frequency signals, improving utilization of pre-training data; and (2) a Mixture-of-Prompts-enabled Transformer (MoPFormer) that uses dataset-specific prompts to capture heterogeneous temporal patterns via a lightweight routing mechanism. SEMPO achieves zero-shot and few-shot forecasting on various datasets using less parameters and pre-training data points. Experiments show that it outperforms several LLM-based models.

**Questions:**

Q1. The claimed advantage of using less pre-training data is not convincingly demonstrated. The paper does not clearly specify how the number of datasets or total data volume used for pre-training compares to other models. Additionally, the number of pre-training epochs significantly affects the effective data utilization and should be reported to ensure a fair and meaningful comparison.

Q2. MoE architectures typically rely on sparse routing to activate only a subset of expert networks, while maintaining diversity among the experts is essential for generalization. It remains unclear how SEMPO addresses these challenges, particularly how it ensures both routing efficiency and functional diversity in the proposed prompt-based expert system.

Q3. Please respond to the other concerns raised in W1-W4.

**Ethical Concerns:**

["NO or VERY MINOR ethics concerns only"]

**Final Justification:**

The authors have sucessfully addressed my concerns. I would adjust my score accordingly

**Limitations:**

Yes

**Paper Formatting Concerns:**

There are no formatting concerns in this paper.

**Quality:**

3

**Strengths And Weaknesses:**

Strengths:

S1. SEMPO achieves state-of-the-art results in zero-shot and few-shot time series forecasting across 16 datasets, despite using only 6.5M parameters and a modest 83M time points for pre-training, which are substantially smaller than competing models that use hundreds of millions of parameters and billions of data points.

S2. The proposed EASD module effectively captures both high- and low-energy frequency components of time series.

S3. The paper provides extensive experiments on benchmark time series datasets covering various domains. It includes zero-shot, few-shot, ablation, and efficiency analyses that can demonstrate the superiority of SEMPO.

\
Weaknesses:

W1. Some claims are not sufficiently justified. One notable example is the claim that existing time series FMs suffer from energy bias and fail to effectively handle low-energy frequency components is primarily supported by a case study. A more rigorous justification for this limitation by other models would be necessary to convincingly support this claim.

W2. The presentation of SEMPO’s data efficiency is unclear. While the model is positioned as requiring less pre-training data, the paper does not clearly quantify the significance of this reduction compared to baselines. It is unclear how many data points are utilized in other models. Also, the epochs for pre-training play an important role in its real data utilization.

W3. There is a lack of clarity regarding the relationship between the pre-training and inference stages. There’s a gap between training and inference stage. It is better to illustrate the process of this two stages in details.

W4. The analysis of low-energy components in the experiments is weak. The paper includes only a qualitative case study to demonstrate SEMPO’s effectiveness in modeling such signals. However, a more comprehensive quantitative evaluation is preferred to be provided.

---

> ### Author Rebuttal · Authors · 2025-07-31
>
> Thank you very much for the constructive and positive comments on our technical contribution and empirical justification. Please see our response to your comments one by one below.
>
> >**Weakness #1** Lack of sufficient justification for the energy bias issue.
>
> Thank you very much for the suggestion. We’d like to clarify that the claim regarding energy bias has been widely justified in the **task-specific forecasting domain** in the literature [Refs1-3], and we explained the basis of this claim in **Lines 145-148**. This phenomenon reflects a learning bias in Transformers, where the self-attention tends to prioritize low-energy components over low-energy ones. Such limitation has been **rigorously supported** by both empirical and theoretical studies. For example, Fredformer [Ref1] and FilterNet [Ref2] provide comprehensive empirical evidence showing that the dominance of low-frequency components, due to their **substantial spectral energy**, can lead to frequency bias, as Transformer-based forecasting models tend to disproportionately attend to these signals during learning; Amplifier [Ref3] complements this by offering a theoretical justification that frequency components with low spectral energy contribute minimally to the training loss and receive slower parameter updates compared to high-energy components.
>
> Building upon these previous findings, and given that most existing foundation models (FMs) [Refs4-6] are built upon Transformer architectures, we empirically investigate whether energy bias persists in **general-purpose forecasting domain**. Our experiments on ETTh1 and Electricity datasets confirm its presence in foundation models, such as Chronos$_S$ [Ref6] and Moirai$_L$ [Ref5], as illustrated in **Figures 1, 4**. To further address your concern regarding this claim, we have added additional quantitative results in our response to **Weakness #4**, showing that other FMs like TimesFM and Timer also struggle to effectively capture low-energy frequency components.
>
> Together, prior studies and our quantitative and qualitative results provide a rigorous justification for our claim that existing FMs exhibit energy bias.
>
> > **Weakness #2 and Question #1** Unclear presentation of SEMPO’s data efficiency, such as pre-training data size and epochs.
>
> We’d like to clarify that both the **model size and pre-training data size** of SEMPO and all baseline FMs have been reported in **Table 1 in main text and Table 6 in Appendix C.2** (in the $(\cdot,\cdot)$ values just below the names of the models). SEMPO achieves strong zero-shot forecasting performance with just 83M time points, underscoring its remarkable data efficiency compared to baselines trained on data with hundreds of millions of time points to billions (or even hundreds of billions) of time points.
>
> Additionally, the number of pre-training epochs have been provided in **Lines 1010-1012 in Appendix C.3**: in SEMPO, the energy-aware pre-training is conducted for 10 epochs, followed by 20 epochs of MoP tuning. To enable a straightforward comparison, Table A1 summarizes the pre-training epochs and data sizes of SEMPO and other four strong FMs. Notably, SEMPO requires only 83M time points,  with 30 epochs of training, achieving a substantial reduction in training data and demonstrating superior data efficiency compared to baseline FMs trained on billions of time points.
>
> ```
> Table. A1. Pre-training epochs of SEMPO and four representative FMs.
> ```
> |Models|SEMPO|TTM|Timer|Chronos|Moment|
> |-|-|-|-|-|-|
> |Data size|83M|1B|28B|84B|1.13B|
> |Epoch|30|20|10|200|2|
>
> To further validate SEMPO’s data efficiency, we extended the pre-training by 5 additional epochs in either the pre-training or the tuning stage, with the results presented in Table A2 below. TTM with 20 training epochs is used as baseline. SEMPO reaches optimal performance within 30 total epochs under 83M time points, indicating that SEMPO has effectively converged and fully utilized the pre-training data.
>
> ```
> Table. A2. Zero-shot performance with increased pre-training epochs, averaged over horizons of {96,192,336,720}.
> ```
> |Datasets|ETTh2||ETTm2||Electricity||Traffic||
> |-|-|-|-|-|-|-|-|-|
> |Metrics|MSE|MAE|MSE|MAE|MSE|MAE|MSE|MAE|
> |SEMPO (10,20)|**0.342**|**0.391**|**0.286**|0.342|**0.197**|**0.295**|**0.467**|**0.344**|
> |SEMPO (15,20)|0.348|0.396|0.288|0.340|0.206|0.304|0.497|0.363|
> |SEMPO (10,25)|0.349|0.398|0.287|0.340|0.202|0.300|0.483|0.353|
> |TTM (20)|0.357|0.396|0.288|**0.330**|0.215|0.303|0.559|0.365|
>
> > **Weakness #3** Unclear relation between pre-training and inference stages.
>
> We’d like to clarify that we’ve provided the **relationships/differences** for the energy-aware pre-training stage, the MoP Tuning stage, and the inference stage in **Figure 2 (Details) in Appendix B**.
> As mentioned in **Lines 867–881 of Appendix B.1**, the energy-aware pre-training does not include the MoP module, which is introduced and used only in the MoP tuning.
>
> As also discussed in **Lines 886–902 of Appendix B.2**, after pre-training, the model enters the fine-tuning phase, where the model is trained on the target dataset using the same joint optimization strategy as in the MoP tuning. In this stage, only the parameters in the mixture of prompts and prediction heads are updated, while the Transformer backbone remains frozen. During zero-shot inference, the reconstruction decoder and head are removed, and there are no parameter updates in all modules.  We will clarify these relationships and differences in the final version.
>
> > **Weakness #4** Weak analysis of low-energy components.
>
> We appreciate your constructive suggestion. We’d like to clarify that our analysis of low-energy components is not limited to the qualitative case study in Figure 4. In fact, we’ve also investigated SEMPO’s capability to model low-energy signals across **diverse prediction lengths**, with the results shown in **Figure 7 in Appendix D.1**.
>
> In terms of quantitative analysis, we conducted ablation studies comparing SEMPO with its variant **A.1 Multi-band Masking**, which removes the energy-based partitioning mechanism (i.e., a variant that also ignores the energy bias, as what is done in existing FMs). The results across **all** datasets and forecasting horizons in Table 3 demonstrate the superior performance of SEMPO over this variant, in which modeling low-energy components is the only difference between SEMPO and this variant. The same observation can be found in the full ablation study results in **Table 13 in Appendix D.6**.
>
> While these results provide meaningful evidence, we agree that more quantitative evaluations would strengthen the argument. In response, we justified the importance of low-energy modeling from another perspective, i.e., whether including such a component in the pre-training could improve the performance of existing FMs.  The results of SEMPO and three strong FMs under two settings: with and without low-energy modeling are given in **Table A3** below. Note that as the energy distribution varies across time windows and **the distinction between high- and low-energy components is inherently relative**, applying a unified threshold for statistical evaluation across the entire dataset is infeasible. We therefore randomly selected five batch samples from Traffic and ETTm1 and report their averaged results in **Table A3**.
>
> ```
> Table A3. Impact of modeling low-energy components (LEC) on MAE across different FMs.
> ```
> |Datasets|LEC Setting|Chronos$_L$|TimesFM|Timer|SEMPO|
> |-|-|-|-|-|-|
> |Traffic|w/o LEC|0.381|-|0.391|0.304|
> ||with LEC|0.341|-|0.357|**0.256**|
> ||Improvement|10.5%|-|8.7%|**15.8%**|
> |ETTm1|w/o LEC|0.502|0.450|0.491|0.472|
> ||with LEC|0.455|**0.394**|0.434|0.409|
> ||Improvement|9.4%|12.4%|9.6%|**13.3%**|
>
> Two key observations can be found in **Table A3**. 1) Adding the low-energy modeling consistently results in very significantly decreased MAE across all FMs, highlighting the positive role these components play in enhancing the prediction accuracy; 2) SEMPO achieves the largest performance gains from incorporating low-energy components, with a 13.3% improvement on ETTm1 and 15.8% on Traffic, indicating its superior ability to effectively mitigate energy bias and model low-energy signals. We will add this quantitative evaluation to the final version.
>
> > **Question #2** How it ensures routing efficiency and functional diversity？
>
> Unlike traditional MoE that requires sparse activation over large expert networks to ensure routing efficiency and reduce computation, SEMPO adopts **lightweight, learnable prompt vectors** as experts. Based on these lightweight experts, SEMPO applies soft gating via a simple linear-softmax function over input tokens, enabling efficient expert selection without the need for sparse activation. These prompts are prepended to the key-value pairs in self-attention, allowing all tokens to share the same Transformer backbone. This avoids multi-branch execution, resulting in minimal computational overhead only, and well supports parameter-efficient adaptation.
>
> To ensure functional diversity, each prompt-based expert is tailored to a specific small dataset, independently learned, and assigned input-adaptive weights via soft gating. A reparameterization strategy further transforms each expert into structured key-value pairs through a two-layer MLP, enhancing its expressiveness and promoting diverse functional roles across the expert set.
>
> >**References**:
> - [Ref1] Fredformer: Frequency Debiased Transformer for Time Series Forecasting, KDD 2024
> - [Ref2] FilterNet: Harnessing Frequency Filters for Time Series Forecasting, NeurIPS 2024.
> - [Ref3] Amplifier: Bringing Attention to Neglected Low-Energy Components in Time Series Forecasting, AAAI 2025.
> - [Ref4] Timer: Generative Pre-trained Transformers Are Large Time Series Models, ICML 2024
> - [Ref5] Unified training of universal time series forecasting transformers, ICML 2024
> - [Ref6] Chronos: Learning the language of time series, TMLR 2024

---

> > ### Comment · Reviewer_uD87 · 2025-08-04
> >
> > The authors have addressed my concerns. I will adjust my score accordingly.

---

> > > ### Author Response · Authors · 2025-08-04
> > > **Response to Reviewer uD87**
> > >
> > > Dear Reviewer uD87,
> > >
> > > We sincerely appreciate your consideration of our responses and the adjustment to your recommendation score. We are very pleased that our reply has addressed your concerns. If you have any further questions or concerns, please feel free to let us know.
> > >
> > > Best regards,
> > >
> > > Authors

---

### Official Review · Reviewer_3Jvu · 2025-06-27

**Clarity:** 4
**Significance:** 3
**Originality:** 3
**Rating:** 5
**Confidence:** 4

**Summary:**

This paper tackles the time series forecasting task by utilising a lightweight foundation model, involving 2 key modules, one decomposes the time series in the spectral domain by thresholding the frequencies and masking the frequencies randomly; and one mixture of prompts block. It is claimed that with such modules the model improves the time series forecasting performance in MSE, MAE or sMAPE.

**Questions:**

- What do Hec and Lec mean? What would you expect, if the energy-wise spectral partitioning leads to more than 2 partitions?

- In figure 4, what is "input", and what is the point of plotting "input" in the figure?

- In figure 8a in Appendix, could you explain the peak at N_m=5? Does it make sense to show the cases with N_m > 6?

**Ethical Concerns:**

["NO or VERY MINOR ethics concerns only"]

**Final Justification:**

My questions are mostly answered.

**Limitations:**

Although claimed in the checklist, I failed to find a proper discussion of limitations in the conclusion.

**Paper Formatting Concerns:**

I don't see any formatting issue.

**Quality:**

3

**Strengths And Weaknesses:**

- Strength

  The paper is well written and easy to follow. The approach is clearly described, and the experimental setup and the results are well illustrated.

  The proposed approach is to the best of my knowledge novel and sound.

- Weakness

  It seems the results in the experiments are from only one run. Since the masks are generated randomly, it would be more convincing if multiple runs are conducted and the standard deviations are reported.

  It is claimed in the checklist 7, but I failed to see a discussion about the statistical test in the paper.

---

> ### Author Rebuttal · Authors · 2025-07-31
>
> Thank you very much for the constructive comments. We are grateful for the positive comments on our methodological innovation, paper clarity and empirical justification. Please see our response to your comments one by one below.
>
> >**Weakness #1** It seems the results in the experiments are from only one run. Since the masks are generated randomly, it would be more convincing if multiple runs are conducted and the standard deviations are reported
>
> All experimental results are obtained by averaging over three runs with different random seeds. In deep time series forecasting, performance variance across seeds is typically small, and following prior work [Refs1-3], we report only the mean values. To address your concern regarding random sampling, we additionally report the standard deviation of the 5\% few-shot experiments to demonstrate the stability of our method in **Table A1** below.
>
> ```
> Table. A1. Statistical results (mean ± std) of SEMPO and TTM under the 5% few-shot setting. The horizon length H is all set to 336.
> ```
> |Models|SEMPO||TTM||
> |---|---|---|---|---|
> |Metrics|MSE|MAE|MSE|MAE|
> |ETTh1|0.425 $\pm$ 0.0006|0.439 $\pm$ 0.0000|**0.400 $\pm$ 0.0010**|**0.421 $\pm$ 0.0006**|
> |ETTh2|**0.355 $\pm$ 0.0005**|**0.402 $\pm$ 0.0000**|0.383 $\pm$ 0.0012|0.415 $\pm$ 0.0008|
> |ETTm1|**0.376 $\pm$ 0.0012**|**0.393 $\pm$ 0.0005**|0.389 $\pm$ 0.0005|0.396 $\pm$ 0.0008|
> |ETTm2|**0.274 $\pm$ 0.0000**|**0.328 $\pm$ 0.0000**|0.319 $\pm$ 0.0008|0.347 $\pm$ 0.0000|
> |Weather|**0.249 $\pm$ 0.0005**|**0.285 $\pm$ 0.0000**|0.255 $\pm$ 0.0000|0.292 $\pm$ 0.0000|
> |Electricity|**0.168 $\pm$ 0.0000**|**0.267 $\pm$ 0.0000**|0.185 $\pm$ 0.0005|0.282 $\pm$ 0.0005|
> |Traffic|**0.421 $\pm$ 0.0008**|**0.292 $\pm$ 0.0000**|0.446 $\pm$ 0.0000|0.315 $\pm$ 0.0000|
>
> The results show that both SEMPO and TTM exhibit stable performance across different seeds. Furthermore, SEMPO demonstrates its superiority by outperforming TTM in both stability and predictive accuracy in 12 out of 14 cases.
>
> >**Question #1** What do Hec and Lec mean? What would you expect, if the energy-wise spectral partitioning leads to more than 2 partitions?
>
> Hec refers to high-energy components, and Lec refers to low-energy components. If energy-wise spectral partitioning yields more than two groups, it effectively creates multiple patches along the energy axis.  This is similar to the frequency patching technique. The latter offers finer granularity to capture energy patterns but introduces significant computational overhead during masking.
>
> >**Question #2** In figure 4, what is "input", and what is the point of plotting "input" in the figure?
>
> In **Figure 4**, "input" refers to the spectrum of the lookback window $X_{1:L}$. In time series forecasting, it is a common practice to include the lookback window in both time- and frequency-domain visualizations to present a complete forecasting sample, see similar examples in [Ref1, Ref4]. More importantly, it provides essential contextual information that helps assess whether the model's predicted spectrum preserves the spectral structure of the input and whether the model tends to emphasize certain frequency components.
>
> >**Question #3** In figure 8a in Appendix, could you explain the peak at $N_m=5$? Does it make sense to show the cases with $N_m>6$?
>
> In principle, increasing $N_m$ provides the model with more complementary frequency perspectives via multiple random samplings, facilitating the learning of diverse spectral features. However, the spike observed at $N_m=5$ is likely caused by stochastic noise rather than a stable trend. Such fluctuations may arise as redundant or conflicting frequency views introduce noise or disrupt convergence, ultimately leading to performance degradation.
>
> Unfortunately, it does not make sense to show cases with $N_m>6$. This is because, as shown in **Figure 8(a)**, performance does not consistently improve beyond $N_m=4$, and it may even degrade due to increased randomness in masking and potential over-fragmentation of frequency views. Meanwhile, a larger $N_m$ imposes considerable memory and computational overhead, as frequency masking is applied separately to $N_m$ duplicated inputs in both high- and low-energy branches. Considering the trade-off between performance and efficiency, we choose not to explore $N_m>6$.
>
> >**Limitation** Although claimed in the checklist, I failed to find a proper discussion of limitations in the conclusion.
>
> We briefly touched on this aspect in **Lines 360-362**, where we discussed future directions that reflect the current limitations of SEMPO. Specifically, one limitation is that the channel-independent mechanism may hinder the modeling of critical multivariate interactions. Another is that supporting only point forecasting reduces the SEMPO’s flexibility in capturing uncertainty through probabilistic forecasting. These limitations may constrain the SEMPO’s ability to capture richer time series characteristics.
>
> We will add all the above empirical results and clarifications in the final version and hope we have addressed your concerns.
>
> >**References**:
> - [Ref1] Time-MoE: Billion-Scale Time Series Foundation Models with Mixture of Experts, ICLR 2025
>
> - [Ref2] Tiny Time Mixers (TTMs): Fast Pre-trained Models for Enhanced Zero/Few-shot Forecasting of Multivariate Time Series, NeurIPS 2024
>
> - [Ref3] Timer: Generative Pre-trained Transformers Are Large Time Series Models, ICML 2024
>
> - [Ref4] Fredformer: Frequency Debiased Transformer for Time Series Forecasting, KDD 2024

---

> > ### Comment · Reviewer_3Jvu · 2025-08-01
> > **Reply**
> >
> > Thanks to the authors for the answer.
> >
> > Regarding weakness 1, is it stated anywhere in the paper ```averaging over three runs```? Because I failed to find it.
> >
> > And thanks for the table A1. One follow-up question, is it conducted with the same setting as table 2 in the paper? Why the average values are different from table 2?

---

> > > ### Author Response · Authors · 2025-08-01
> > > **Response to Reviewer 3Jvu**
> > >
> > > Dear Reviewer 3Jvu,
> > >
> > > We greatly appreciate your continuous valuable feedback. Please find our response to your remaining concerns below.
> > >
> > > >**Follow-up question #1** Regarding weakness 1, is it stated anywhere in the paper averaging over three runs? Because I failed to find it.
> > >
> > > This information was indeed not stated in the paper. To address your concern, we will clearly clarify this point in the final version.
> > >
> > > >**Follow-up question #2** And thanks for the table A1. One follow-up question, is it conducted with the same setting as table 2 in the paper? Why the average values are different from table 2?
> > >
> > > The settings for the two set of results are different. Table 2 reports results averaged over four forecasting horizons $H \in \\{96,192,336,720\\}$, while Table A1 presents results averaged over different random seeds under a single horizon $H=336$. The values in Table A1 are **consistent** with those in the full version of Table 2, namely **Table 11 in Appendix D.6**, at the same horizon.
> > >
> > > We sincerely hope the above replies help address your follow-up questions. If you have any further questions or concerns, please feel free to let us know.
> > >
> > > Best regards,
> > >
> > > Authors

---

### Official Review · Reviewer_Y38u · 2025-07-02

**Clarity:** 3
**Significance:** 3
**Originality:** 2
**Rating:** 4
**Confidence:** 4

**Summary:**

The paper proposes SEMPO, a lightweight foundation model designed for general time series forecasting that significantly reduces both model size and pre-training data requirements, while still achieving state-of-the-art performance in zero-shot and few-shot scenarios.

**Questions:**

In lines 126–130, the paper introduces the concepts of $D_tune$ and $D_test$ in the context of few-shot and zero-shot evaluation. However, it is unclear whether these datasets are drawn from entirely new domains or from domains already included in the pre-training set $T_train$. Please clarify this distinction to help assess the generalization capabilities of the proposed model more accurately.

	In Figure 1 (right), SEMPO (Zero) and SEMPO (Few) are marked separately, while other models such as PatchTST and Time-LLM are not differentiated by evaluation setting. This inconsistency may lead to confusion when interpreting the results. It would be clearer to present two separate figures: one for zero-shot comparisons and another for few-shot comparisons, ensuring a fair and consistent visual comparison across models.

**Ethical Concerns:**

["NO or VERY MINOR ethics concerns only"]

**Final Justification:**

I would like to raise my score as my concerns are mostly addressed, although I am still have reservation on novelty.

**Limitations:**

yes

**Quality:**

3

**Strengths And Weaknesses:**

Strength

1.	The paper is well-structured, with clear organization and a logical flow that makes it easy to follow.

2.	Contribution: The paper explores an efficient way to address time series forecasting through Energy-aware Spectral Decomposition and Mixture-of-Prompts Transformer. Extensive experiment results demonstrate that SEMPO is able to outperform baselines with only 6.5M parameters which is pre-trained on 83 million time points

Weakness

1.	The core contribution of SEMPO is to present a lightweight yet effective foundation model for time series forecasting. However, the experimental evaluation primarily focuses on zero-shot and few-shot settings, where many of the baseline methods were not originally designed to perform. This may limit the fairness and comprehensiveness of the comparison. It would strengthen the empirical analysis if the authors could additionally include in-distribution evaluations—similar to what was done in Time-MoE—to demonstrate that SEMPO also achieves competitive or superior performance in standard supervised settings, despite its reduced model size.

2.	In lines 51–53, the authors state: “Specifically, modeling full energy spectrum of temporal signals from pre-training data is essential for learning a comprehensive knowledge base of temporal patterns to underpin strong transferability across different datasets.” This is an important claim, but it lacks empirical evidence or citations to support it.


3.	While the proposed Mixture-of-Prompts (MoP) mechanism contributes to model adaptability, the overall technical novelty of SEMPO is limited. The use of FFT for spectral decomposition is a well-established technique in time series analysis, and the design of the model architecture—particularly the encoder-decoder structure and masking strategies—follows relatively standard practices. As a result, the contribution of the model as a whole may not be sufficient for a top-tier venue.

4.	There are several formatting issues in the current draft, such as broken links to corresponding figures and missing or improperly linked citations. These issues hinder readability and should be addressed to improve the overall presentation quality.

---

> ### Author Rebuttal · Authors · 2025-07-31
>
> We sincerely appreciate your constructive and positive comments on the clear presentation and technical contribution. Please see our response to your comments one by one below.
>
> >**Weakness #1** Lack of in-distribution evaluations similar to Time-MoE
>
> Thank you very much for the insightful suggestion. To ensure fairness and enhance the comprehensiveness of our evaluation, we have added in-distribution experiments under the standard supervised setting adopted by Time-MoE. Specifically, we compare SEMPO against LLM-based time series foundation models (FMs) (S2IP-LLM and GPT4TS) as well as strong task-specific models (iTransformer, DLinear, PatchTST and TimesNet). Results on four datasets of different scales are presented in **Table A1** below:
>
> ```
> Table A1. In-domain forecasting results on four datasets.
> ```
> |Dataset |Horizon|SEMPO||GPT4TS||S2PLLM||iTransformer||DLinear||PatchTST||TimesNet||
> |---|---|---|---|---|---|---|---|---|---|---|---|---|---|---|---|
> |||MSE|MAE|MSE|MAE|MSE|MAE|MSE|MAE|MSE|MAE|MSE|MAE|MSE|MAE|
> |ETTh2|96|**0.273**|**0.334**|0.285|0.342|0.278|0.340|0.297|0.348|0.302|0.368|0.274|0.337|0.351|0.399|
> ||192|0.333|**0.376**|0.354|0.389|**0.246**|0.385|0.371|0.403|0.404|0.433|0.341|0.382|0.394|0.429|
> ||336|0.355|0.400|0.373|0.407|0.367|0.406|0.404|0.428|0.511|0.498|**0.329**|**0.384**|0.415|0.443|
> ||720|0.399|0.435|0.406|0.441|0.400|0.436|0.424|0.444|0.815|0.640|**0.379**|**0.422**|0.477|0.481|
> |ETTm2|96|**0.160**|**0.251**|0.173|0.262|0.165|0.257|0.175|0.266|0.164|0.255|0.166|0.256|0.233|0.305|
> ||192|**0.221**|**0.294**|0.229|0.301|0.222|0.299|0.242|0.312|0.224|0.304|0.223|0.296|0.265|0.328|
> ||336|**0.273**|**0.328**|0.286|0.341|0.277|0.330|0.282|0.340|0.277|0.339|0.274|0.329|0.379|0.392|
> ||720|**0.349**|**0.380**|0.378|0.401|0.363|0.390|0.378|0.398|0.371|0.401|0.362|0.385|0.390|0.407|
> |Weather|96|**0.143**|**0.193**|0.162|0.212|0.145|0.195|0.159|0.208|0.170|0.230|0.149|0.198|0.193|0.244|
> ||192|**0.189**|**0.234**|0.204|0.248|0.190|0.235|0.200|0.248|0.212|0.267|0.194|0.241|0.320|0.329|
> ||336|**0.240**|**0.280**|0.254|0.286|0.243|0.280|0.253|0.289|0.257|0.305|0.245|0.282|0.363|0.366|
> ||720|0.325|0.338|0.326|0.337|**0.312**|**0.326**|0.321|0.338|0.318|0.356|0.314|0.334|0.440|0.404|
> |Traffic|96|**0.355**|**0.246**|0.388|0.282|0.379|0.274|0.363|0.265|0.411|0.294|0.360|0.249|0.611|0.323|
> ||192|**0.373**|**0.253**|0.407|0.290|0.397|0.282|0.385|0.273|0.421|0.298|0.379|0.256|0.609|0.327|
> ||336|**0.384**|**0.260**|0.412|0.294|0.407|0.289|0.396|0.277|0.431|0.304|0.392|0.264|0.616|0.335|
> ||720|**0.427**|**0.286**|0.450|0.312|0.440|0.301|0.445|0.312|0.468|0.325|0.432|**0.286**|0.656|0.349|
> |**1$^{st}$ Count**||**25**||0||3||0||0||5||0||
>
> The results show that SEMPO consistently outperforms six advanced time series models, demonstrating that its pre-training framework effectively equips the model with strong inductive biases for downstream adaptation, despite its lightweight design. These results will be included in the final version to address this concern.
>
> >**Weakness #2** Lack empirical evidence or citations to support claim on ''modeling full energy spectrum of temporal signals from pre-training data...''
>
> Low-energy frequency components represent subtle yet informative short-term variations, such as periodicities over short durations, which serve as good indicators for forecasting. Prior studies [Refs1-3] have revealed that attending to these low-frequency components with substantial spectral energy and mitigating this energy bias are critical for enhancing performance in **task-specific forecasting domain**. In particular, Fredformer [Ref1] and FilterNet [Ref2] provide **compelling empirical evidence** that the dominance of low-frequency components with substantial spectral energy can create a bottleneck to full-spectrum information utilization, ultimately hindering forecasting performance.
>
> Building upon this, multi-source time series pre-training can be viewed as an **aggregation** of learning processes across multiple individual datasets. If energy bias occurs within each dataset, it can compound during pre-training, leading to irreversible information loss. Therefore, we extend these prior findings to **general-purpose forecasting domain**, arguing that modeling the full energy spectrum is also essential for learning comprehensive forecasting patterns from a large collection of pre-training datasets to underpin strong generalization across datasets.
>
> To further address your concern regarding empirical evidence, we conducted experiments comparing SEMPO with three strong FMs (Chronos$_L$, TimesFM, and Timer) under two settings: with and without low-energy components, to highlight the indispensable role of low-energy pattern modeling, please see our response to **Reviewer uD87**’s **Weakness #4** for the detailed results and discussion. We will include the references and empirical evidence in the final version.
>
> >**Weakness #3** The use of FFT and the encoder-decoder architecture with masking strategies appears standard and lacks novelty
>
> FFT-based spectral decomposition is indeed widely adopted in time series analysis, such as in [Refs1-4]. However, it is important to clarify that these prior works primarily address the **learning bias between high- and low-frequency components**, a phenomenon characterized by the prioritization of low-frequency features at the expense of high-frequency features.
> In contrast, our work goes a step further by empirically revealing that existing time series FMs [Refs5-6] are generally biased toward high-energy components, while neglecting low-energy components that may distribute across mid-, high-, or even low-frequency bands (**see Figures 1, 4, 7**). The coarse-grained frequency bias correction is insufficient in this context, as it can lead to significant information loss of low-energy frequency information. To address this, we propose the novel EASD module, which performs adaptive spectral partitioning and dual-branch spectral masking to model low-energy yet informative frequency signals that are ignored in current FMs.
>
> Additionally, most existing FMs adopt Transformer-based architectures, such as Moirai [Ref5] and Chronos [Ref6]. Leveraging a well-established encoder-decoder framework and following standard design practices does **not diminish** the novelty of our work in the two proposed modules. Instead, it enables us to clearly identify the above **critical limitation** in current Transformer-based FMs, namely **energy bias**. Building upon this mature architecture, we further propose the novel MoPFormer module, which enables parameter-efficient model adaptation across different datasets and domains. Together, these two innovative modules enable SEMPO to substantially reduce both the pre-training data scale and model size, while achieving strong generalization.
>
> >**Weakness #4** Several formatting issues in the current draft
>
> Thank you for pointing this out. The broken links may be due to the separation between the main text and appendix. We will address the formatting issues, including figure links and citations, in the final version to ensure preferred readability.
>
> >**Question #1** Clarify whether $D_{tune}$ and $D_{test}$ are drawn from new domains or overlap with the pre-training set $T_{train}$
>
> Following standard few-shot and zero-shot settings, $D_{tune}$ and $D_{test}$ are drawn from entirely new domains that are not presented in $T_{train}$. We will clarify this point in our final version.
>
> >**Question #2** Fair and consistent visual comparison in Figure 1 (right)
>
> We would like to clarify that Time-LLM, GPT4TS, and PatchTST are used only in the few-shot setting, while other FMs are evaluated in the zero-shot setting (see **Table 1 and Table 2**). Accordingly, we distinguish SEMPO (Zero) and SEMPO (Few) for fair comparison in each setting.
>
> To address your concern regarding consistency, we will revise the figure and split it into two subplots for zero- and few-shot settings respectively in the final version.
>
> >**References**:
> - [Ref1] Fredformer: Frequency Debiased Transformer for Time Series Forecasting, KDD 2024
>
> - [Ref2] CATCH: Channel-aware Multivariate Time Series Anomaly Detection via Frequency Patching, ICLR 2025
>
> - [Ref3] Towards a General Time Series Forecasting Model with Unified Representation and Adaptive Transfer, ICML 2025
>
> - [Ref4] FilterNet: Harnessing Frequency Filters for Time Series Forecasting, NeurIPS 2024
>
> - [Ref5] Unified training of universal time series forecasting transformers, ICML 2024
>
> - [Ref6] Chronos: Learning the language of time series, TMLR 2024

---

> > ### Author Response · Authors · 2025-08-05
> > **Hope for your reply**
> >
> > Dear Reviewer Y38u,
> >
> > As the End of the author-reviewer discussion period approaches, we would like to kindly check if our responses have addressed your concerns. Your feedback is truly valuable to us, and we would be more than happy to engage in further discussion and paper improvements. Thank you again for your time and effort in helping us improve our paper!
> >
> > Best regards,
> >
> > Authors

---

### Note · Authors · 2025-08-14

Dear Area Chairs and Reviewers,

We wish to express our sincerest gratitude for your insightful feedback and valuable engagement throughout the review process, which are very helpful for us to improve our paper.

The reviewers generally hold positive opinions of our paper, in that our model offers "**high resource efficiency and practical value**" (Reviewers Y38u, EgfG), our idea is "**novel and sound to the best of his/her knowledge**" (Reviewer 3Jvu), our experiment is "**extensive and thorough**" (Reviewers uD87, EgfG), and our writing is "**well-structured, well-written, and easy to follow**" (Reviewers Y38u, 3Jvu, EgfG).

The reviewers also raised insightful and constructive concerns, and we provided new experiments and analyses to address these concerns, including:

- **Clarification of Technical Novelty (Reviewer Y38u)**: We distinguished SEMPO from earlier spectral decomposition approaches and argued that using well-established architecture and design practices does not diminish our contribution. Crucially, the two proposed modules, EASD and MoPFormer, are both innovative and effective, enabling substantial reductions in pre-training data scale and model size while achieving strong generalization.

- **Claim Justification (Reviewers Y38u, uD87)**: We added further justification to support our claims, along with references and empirical results, demonstrating the indispensable role of low-energy pattern modeling.

- **Expanded Evaluations (Reviewers Y38u, uD87, EgfG, 3Jvu)**: We added all requested evaluations, including full-shot experiments, frequency debiasing baseline Fredformer, latest foundation model baselines Chronos-Bolt and TimesFM (500M), efficiency comparison with TTM, analysis of increased pre-training epochs, and standard deviations across multiple runs.

- **Description of Methodological Details (Reviewers Y38u, uD87, EgfG)**: We clarified the two-stage pre-training and inference pipeline, the differences between $D_{tune},D_{test}$ and $T_{train}$, and the data efficiency.

We are pleased that our response fulfilled reviewers’ expectations and that our efforts to clarify the work were well received, as indicated by the reviewers during the discussion. We are fully committed to integrating all these new results, detailed analyses, and clarifications into the final version of our paper.

Thanks again to AC and the reviewers for your time and effort on our paper.

Best regards,

Authors of Paper #15850

---

### Decision · Program_Chairs · 2025-09-17

**Decision:**

Accept (poster)

**Comment:**

This paper proposes a new lightweight time series foundation model with strong experimental results across 16 datasets in 0 and few short forecasting tasks. The reviewers mostly found the proposed approach to be novel and the experiments to be good although (prior to the author/reviewer discussion completing) not thorough enough. However, after the author/reviewer discussion, the bulk of reviewer concerns were addressed, and the reviewers unanimously favored acceptance for this paper (3 accept, 1 borderline accept). I am recommending acceptance for this paper as a result.